# Monotonic Improvement Guarantees under Non-stationarity for Decentralized PPO

## Abstract

We present a new monotonic improvement guarantee for optimizing decentralized policies in cooperative Multi-Agent Reinforcement Learning (MARL), which holds even when the transition dynamics are non-stationary. This new analysis provides a theoretical understanding of the strong performance of two recent actor-critic methods for MARL, i.e., Independent Proximal Policy Optimization (IPPO) (Schröder de Witt et al., 2020) and Multi-Agent PPO (MAPPO) (Yu et al., 2021), which both rely on *independent ratios*, i.e., computing probability ratios separately for each agent's policy. We show that, despite the non-stationarity that independent ratios cause, a monotonic improvement guarantee still arises as a result of enforcing the trust region constraint over joint policies. We also show this trust region constraint can be effectively enforced in a principled way by bounding independent ratios based on the number of agents in training, providing a theoretical foundation for proximal ratio clipping. Moreover, we show that the surrogate objectives optimized in IPPO and MAPPO are essentially equivalent when their critics converge to a fixed point. Finally, our empirical results support the hypothesis that the strong performance of IPPO and MAPPO is a direct result of enforcing such a trust region constraint via clipping in centralized training, and the good values of the hyperparameters for this enforcement are highly sensitive to the number of agents, as predicted by our theoretical analysis.

## 1 Introduction

In cooperative multi-agent reinforcement learning (MARL), a team of agents must coordinate their behavior to maximize a single cumulative return (Panait & Luke, 2005). In such a setting, partial observability and/or communication constraints necessitate the learning of decentralized policies that condition only on the local action-observation history of each agent. In a simulated or laboratory setting, decentralized policies can often be learned in a centralized fashion, i.e., Centralized Training with Decentralized Execution (CTDE)(Oliehoek & Amato, 2016), which allows agents to access each other's observations and unobservable extra state information during training.

Actor-critic algorithms (Konda & Tsitsiklis, 2000) are a natural approach to CTDE because critics can exploit centralized training by conditioning on extra information not available to the decentralized policies (Lowe et al., 2017; Foerster et al., 2017). Unfortunately, such actor-critic methods have long been outperformed by value-based methods such as QMIX (Rashid et al., 2018) on MARL benchmark tasks such as Starcraft Multi-Agent Challenge (SMAC) (Samvelyan et al., 2019). However, two recent actor-critic algorithms (Schröder de Witt et al., 2020; Yu et al., 2021) have upended this ranking by outperforming previously dominant MARL methods, such as MADDPG (Lowe et al., 2017) and value-decomposed $Q$-learning (Sunehag et al., 2017; Rashid et al., 2018). Both algorithms are multi-agent extensions of Proximal Policy Optimization (PPO) (Schulman et al., 2017) but one uses decentralized critics, i.e., independent PPO (IPPO) (Schröder de Witt et al., 2020), and the other uses centralized critics, i.e., multi-agent PPO (MAPPO) (Yu et al., 2021).

One key feature of PPO-based methods is the use of ratios (between the policy probabilities before and after updating) in the objective. Both IPPO and MAPPO extend this feature of PPO to the multi-agent setting by computing ratios separately for each agent's policy during training, which we call *independent ratios*. Unfortunately, until now there has been no theoretical justification for the use of such independent ratios. In this paper we show that the analysis that underpins the monotonic

policy improvement guarantee for PPO (Schulman et al., 2015) does not carry over to the use of independent ratios in IPPO and MAPPO. Instead, a direct application of this analysis leads to a joint policy optimization and suggests the use of *joint ratios*, i.e., computing ratios between joint policies.

The difference is crucial because, based on the existing trust region analysis for PPO, only a joint ratios approach enjoys a monotonic policy improvement guarantee. Moreover, as independent ratios consider only the change in one agent's policy and ignore the fact that the other agents' policies also change, the transition dynamics underlying these independent ratios are non-stationary (Papoudakis et al., 2019), breaking the assumptions in the monotonic improvement analysis (Schulman et al., 2015). While some studies attempt to extend the monotonic improvement analysis to MARL (Wen et al., 2021; Li & He, 2020), they primarily consider optimizing policies with joint ratios, rather than independent ratios as in our paper, and are thus not applicable to IPPO or MAPPO.

To address this gap, we provide a new monotonic improvement analysis that holds even when the transition dynamics are non-stationary. We show that, despite this non-stationarity, a monotonic improvement guarantee still arises as a result of enforcing the trust region constraint over *joint policies*, i.e., a centralized trust region constraint. In other words, constraining the update of joint policies in centralized training addresses the non-stationarity of learning decentralized policies. This new analysis implies that independent ratios can also enjoy the same performance guarantee as joint ratios if the centralized trust region constraint is properly enforced by bounding independent ratios. In this way both IPPO and MAPPO can guarantee monotonic policy improvement. We also provide a theoretical foundation for proximal ratio clipping by showing that centralized trust region can be enforced in a principled way by bounding independent ratios based on the number of agents in training. Furthermore, we show that the surrogate objectives optimized in IPPO and MAPPO are essentially equivalent when their critics converge to a fixed point.

Finally, we provide empirical results that support the hypothesis that the strong performance of IPPO and MAPPO is a direct result of enforcing such a trust region constraint via clipping in centralized training. Particularly, we show that good values of the hyperparameters for the clipping range are highly sensitive to the number of agents, as these hyperparameters, together with the number of agents, effectively determine the size of the centralized trust region. Moreover, we show that IPPO and MAPPO have comparable performance on SMAC maps with varied difficulty and numbers of agents. This comparable performance also implies that the way of training critics could be less crucial in practice than enforcing a trust region constraint.

## 2 RELATED WORK

The use of trust region optimization in MARL traces back to parameter-sharing TRPO (PS-TRPO) (Gupta et al., 2017), which combines parameter sharing with TRPO for cooperative multi-agent continuous control but provides no theoretical support. Our analysis showing that a trust region constraint is pivotal to guarantee performance improvement in MARL applies to PS-TRPO, among other algorithms. Multi-agent trust region learning (MATRL) (Wen et al., 2021) uses a trust region for independent learning with a game-theoretical analysis in the policy space. MATRL considers independent learning and proposes to enforce a trust region constraint by approximating the stable fixed point via a meta-game. Despite the improvement guarantee for joint policies, solving a meta-game itself can be challenging because its complexity increases exponentially in the number of agents. We instead consider centralized learning and enforce the trust region constraint in a centralized and scalable way. Multi-Agent TRPO (MATRPO) directly extends TRPO to the multi-agent case (Li & He, 2020) and divides the trust region by the number of agents. However, the analysis assumes a private reward for each agent, which yields different theoretical results from ours. Non-stationarity has been discussed in multi-agent mirror descent with trust region decomposition (Li et al., 2021), which first decomposes the trust region for each decentralized policy and then approximates the KL divergence through additional training. However, this method needs to learn a fully centralized action-value function and thus becomes becomes impractical when there are many agents.

## 3 BACKGROUND

**Dec-MDPs.** We consider a *fully cooperative multi-agent task* in which a team of cooperative agents choose sequential actions in a stochastic environment. It can be modeled as a *decentralized Markov*

*decision process* (Dec-MDP), defined by a tuple $\{\mathcal{N}, \mathcal{S}, \mathcal{A}, P, r, \rho_0, \gamma\}$, where $\mathcal{N} \triangleq \{1, \ldots, N\}$ denotes the set of $N$ agents and $\boldsymbol{s} \in \mathcal{S} \triangleq \mathcal{S}^1 \times \mathcal{S}^2 \times \ldots \times \mathcal{S}^N$ describes the joint state of the environment. The initial state $s_0 \sim \rho_0$ is drawn from distribution $\rho_0$, and at each time step $t$, all agents $k \in \mathcal{N}$ choose simultaneously one action $a_t^k \in \mathcal{A}^k$, yielding a joint action $\boldsymbol{a}_t \triangleq a_t^1 \times a_t^2 \times \ldots \times a_t^N \in \mathcal{A} \triangleq \mathcal{A}^1 \times \mathcal{A}^2 \times \ldots \times \mathcal{A}^N$. After executing the joint action $\boldsymbol{a}_t$ in state $\boldsymbol{s}_t$, the next state $\boldsymbol{s}_{t+1} \sim P(\boldsymbol{s}_t, \boldsymbol{a}_t)$ is drawn from transition kernel $P$ and a collaborative reward $r_t = r(\boldsymbol{s}_t, \boldsymbol{a}_t)$ is returned. In a Dec-MDP, each agent $k \in \mathcal{N}$ has a local state $s_t^k \in \mathcal{S}^k$, and chooses its actions with a decentralized policy $a_t^k \sim \pi_k(\cdot|s_t^k)$ based only on its local state. The collaborating team of agents aims to learn a *joint policy*, $\boldsymbol{\pi}(\boldsymbol{a}_t|\boldsymbol{s}_t) \triangleq \prod_{k=1}^N \pi_k(a_t^k|s_t^k)$, that maximizes their expected discounted return, $\eta(\boldsymbol{\pi}) \triangleq \mathbb{E}_{(\boldsymbol{s}_t, \boldsymbol{a}_t)}[\sum_{t=0}^\infty \gamma^t r_t]$, where $\gamma \in [0, 1)$ is a discount factor.

**Policy optimization methods.** For single-agent RL that is modeled as an infinite-horizon discounted Markov decision process (MDP) $\{\mathcal{S}, \mathcal{A}, P, r, \rho_0, \gamma\}$, the performance for a policy $\pi(a|s)$ is defined as: $\eta(\pi) = \mathbb{E}_{(s_t, a_t)}\big[\sum_{t=0}^\infty \gamma^t r(s_t, a_t)\big]$. The action-value function $Q_\pi$ and value function $V_\pi$ are defined as:

$$Q_\pi(s_t, a_t) = \mathbb{E}_{\substack{s_{t+1}\sim p(\cdot|s_t,a_t), \\ a_{t+1}\sim\pi(\cdot|s_{t+1})}}\Big[\sum_{l=0}^\infty \gamma^l r(s_{t+l}, a_{t+l})\Big], \quad V_\pi(s_t) = \mathbb{E}_{a_t\sim\pi(\cdot|s_t)}\Big[Q_\pi(s_t, a_t)\Big].$$

Let the advantage function be $A_\pi(s, a) = Q_\pi(s, a) - V_\pi(s)$; the following useful identity expresses the expected return of another policy $\tilde{\pi}$ in terms of the advantage over $\pi$ (Kakade & Langford, 2002): $\eta(\tilde{\pi}) = \eta(\pi) + \sum_s \rho_{\tilde{\pi}}(s) \sum_a \tilde{\pi}(a|s) A_\pi(s, a)$, where $\rho_{\tilde{\pi}}(s)$ is the state distribution induced by $\tilde{\pi}$. The complex dependency of $\rho_{\tilde{\pi}}(s)$ on $\tilde{\pi}$ makes the righthand side difficult to optimize directly. Schulman et al. (2015) proposed to consider the following surrogate objective

$$L_\pi(\tilde{\pi}) = \eta(\pi) + \sum_s \rho_\pi(s) \sum_a \tilde{\pi}(a|s) A_\pi(s, a) = \eta(\pi) + \mathbb{E}_{(s,a)\sim\rho_\pi}\Big[\frac{\tilde{\pi}(a|s)}{\pi(a|s)} A_\pi(s, a)\Big],$$

where the $\rho_{\tilde{\pi}}$ is replaced with $\rho_\pi$. Define $D_{\text{TV}}^{\max}(\pi, \tilde{\pi}) \triangleq \max_s D_{\text{TV}}\big(\pi(\cdot|s), \tilde{\pi}(\cdot|s)\big)$, where $D_{\text{TV}}$ is the total variation (TV) divergence.

**Theorem 1.** *(Theorem 1 in Schulman et al. (2015)) Let $\alpha = D_{\text{TV}}^{\max}(\pi, \tilde{\pi})$. Then the following bound holds*

$$\eta(\tilde{\pi}) \geq L_\pi(\tilde{\pi}) - \frac{4\epsilon\gamma}{(1-\gamma)^2}\alpha^2,$$

*where $\epsilon = \max_{s,a}|A_\pi(s, a)|$.*

This theorem forms the foundation of policy optimization methods, including Trust Region Policy Optimization (TRPO) (Schulman et al., 2015) and Proximal Policy Optimization (PPO) (Schulman et al., 2017). TRPO suggests a robust way to take large update steps by using a constraint, rather than a penalty, on the TV divergence, and considers the following practical optimization problem,

$$\text{TRPO:} \quad \max_{\tilde{\pi}} \quad \mathbb{E}_{(s,a)\sim\rho_\pi}\Big[\frac{\tilde{\pi}(a|s)}{\pi(a|s)} A_\pi(s, a)\Big], \qquad \text{s.t.} \quad D_{\text{TV}}^{\max}(\pi(\cdot|s), \tilde{\pi}(\cdot|s)) \leq \alpha. \quad (1)$$

This constrained optimization is complicated as it requires using conjugate gradient algorithms with a quadratic approximation to the constraint. PPO simplifies the above optimization by clipping probability ratios $\lambda_{\tilde{\pi}} = \frac{\tilde{\pi}(a|s)}{\pi(a|s)}$ to form a lower bound of $L_\pi(\tilde{\pi})$:

$$\text{PPO:} \quad \max_{\tilde{\pi}} \quad \mathbb{E}_{(s,a)\sim\rho_\pi}\big[\min\big(\lambda_{\tilde{\pi}} A_\pi(s, a), \text{clip}(\lambda_{\tilde{\pi}}, 1-\epsilon, 1+\epsilon) A_\pi(s, a)\big)\big], \quad (2)$$

where $\epsilon$ is a hyperparameter to specify the clipping range.

**Independent PPO and Multi-Agent PPO.** Both IPPO (Schröder de Witt et al., 2020) and MAPPO (Yu et al., 2021) optimize decentralized policies with independent ratios. In particular, the main objective IPPO and MAPPO optimize is

$$\max_{\tilde{\pi}_k} \mathbb{E}_{(s^k, a^k)\sim\rho_{\pi_k}}\big[\min\big(\lambda_{\tilde{\pi}_k} A^k(s^k, a^k), \text{clip}(\lambda_{\tilde{\pi}_k}, 1-\epsilon, 1+\epsilon) A^k(s^k, a^k)\big)\big] \quad \forall k \in \{1, 2, \ldots, N\},$$
$$(3)$$

where $\lambda_{\tilde{\pi}_k} = \frac{\tilde{\pi}_k(a^k|s^k)}{\pi_k(a^k|s^k)}$ denotes the ratio between the decentralized policy probabilities of agent $k$ before and after updating. The difference between IPPO and MAPPO lies in how they estimate the advantage function: IPPO learns a decentralized advantage function $A^k(s^k, a^k) \triangleq \sum_{t=0}^{\infty}[r(s_t^k, a_t^k)] - V(s^k)$ based on the local information $(s^k, a^k)$ for each agent, while MAPPO uses a centralized critic that conditions on centralized state information $s$: $A^k(s^k, a^k) \triangleq \mathbb{E}_{s^{-k}}\left[\sum_{t=0}^{\infty}[r(s_t^k, a_t^k)] - V(\boldsymbol{s})\right]$, where $-k$ refers the set of all agents except agent $k$. Both methods use parameter sharing between agents. Consequently, as all agents share the same actor and critic networks, one can ignore the agent specifier $k$ in the objective and use experience from all agents to update the actor and critic networks:

$$\max_{\tilde{\pi}_\theta} \sum_k \mathbb{E}_{(s^k, a^k) \sim \rho_{\pi_\theta}}\left[\min\left(\lambda_\theta A_\phi(s^k, a^k), \text{clip}(\lambda_\theta, 1 - \epsilon, 1 + \epsilon)A_\phi(s^k, a^k)\right)\right], \qquad (4)$$

where $\lambda_\theta = \frac{\tilde{\pi}_\theta(a^k|s^k)}{\pi_\theta(a^k|s^k)}$, and $\theta$, $\phi$ are shared parameters for policy and advantage networks. The use of independent ratios together with parameter sharing has shown strong empirical results in various MARL benchmark tasks (Schröder de Witt et al., 2020; Yu et al., 2021).

## 4 TRUST REGION ANALYSIS FOR MARL

In this section, we first directly apply TRPO's trust region analysis to cooperative MARL, which yields joint ratios rather than the independent ratios adopted in IPPO and MAPPO. We then show that optimization with independent ratios induces non-stationarity in MARL, which breaks the stationarity assumption in the trust region analysis. Finally, we provide a new analysis that shows how monotonic policy improvement can still arise from non-stationary transition dynamics with independent ratios.

### 4.1 OPTIMIZATION WITH JOINT RATIOS

Consider the joint policy $\boldsymbol{\pi}(\boldsymbol{a}|\boldsymbol{s})$ and the centralized advantage function $A_{\boldsymbol{\pi}}(\boldsymbol{s}, \boldsymbol{a}) = Q_{\boldsymbol{\pi}}(\boldsymbol{s}, \boldsymbol{a}) - V_{\boldsymbol{\pi}}(\boldsymbol{s})$. Then, the trust region analysis for single-agent RL carries over directly to MARL with the surrogate objective as $L_{\boldsymbol{\pi}}(\tilde{\boldsymbol{\pi}}) = \eta(\boldsymbol{\pi}) + \sum_{\boldsymbol{s}} \rho_{\boldsymbol{\pi}}(\boldsymbol{s}) \sum_{\boldsymbol{a}} \tilde{\boldsymbol{\pi}}(\boldsymbol{a}|\boldsymbol{s}) A_{\boldsymbol{\pi}}(\boldsymbol{s}, \boldsymbol{a})$. One can consider the same optimization problem for TRPO shown in equation 1,

$$\max_{\tilde{\boldsymbol{\pi}}} \quad \mathbb{E}_{(\boldsymbol{s}, \boldsymbol{a}) \sim \rho_{\boldsymbol{\pi}}}\left[\frac{\tilde{\boldsymbol{\pi}}(\boldsymbol{a}|\boldsymbol{s})}{\boldsymbol{\pi}(\boldsymbol{a}|\boldsymbol{s})} A_{\boldsymbol{\pi}}(\boldsymbol{s}, \boldsymbol{a})\right], \qquad \text{s.t.} \quad D_{\text{TV}}^{\max}(\boldsymbol{\pi}(\cdot|\boldsymbol{s}), \tilde{\boldsymbol{\pi}}(\cdot|\boldsymbol{s})) \le \alpha. \qquad (5)$$

The trust region constraint is enforced over joint policies, which we refer as a *centralized trust region constraint*. With joint ratios defined as $\boldsymbol{\lambda}_{\tilde{\boldsymbol{\pi}}} = \frac{\tilde{\boldsymbol{\pi}}(\boldsymbol{a}|\boldsymbol{s})}{\boldsymbol{\pi}(\boldsymbol{a}|\boldsymbol{s})} = \prod_{k=1}^N \left[\frac{\tilde{\pi}_k(a^k|s^k)}{\pi_k(a^k|s^k)}\right]$, one can simplify the above optimization as PPO to have the following objective,

$$\text{JR-PPO:} \quad \max_{\tilde{\boldsymbol{\pi}}} \quad \mathbb{E}_{(\boldsymbol{s}, \boldsymbol{a}) \sim \rho_{\boldsymbol{\pi}}}\left[\min\left(\boldsymbol{\lambda}_{\tilde{\boldsymbol{\pi}}} A_{\boldsymbol{\pi}}(\boldsymbol{s}, \boldsymbol{a}), \text{clip}(\boldsymbol{\lambda}_{\tilde{\boldsymbol{\pi}}}, 1 - \epsilon, 1 + \epsilon)A_{\boldsymbol{\pi}}(\boldsymbol{s}, \boldsymbol{a})\right)\right]. \qquad (6)$$

We call the resulting algorithm Joint Ratio PPO (JR-PPO) (see Algorithm 1 in the appendix).

Unlike IPPO and MAPPO, JR-PPO consider joint ratios over the joint policies, rather than independent ones. This difference is crucial, as joint ratios naturally enjoy the monotonic improvement guarantee carried over from the single-agent trust region analysis:, i.e., Theorem 1. Furthermore, the objective used in IPPO and MAPPO is not equivalent to the above objective as they are lower bounds of different objectives. Thus, Theorem 1 does not imply any guarantees for IPPO and MAPPO.

### 4.2 OPTIMIZATION WITH INDEPENDENT RATIOS

Optimization with independent ratios, however, induces non-stationarity in MARL. When optimizing decentralized policies, the environment is non-stationary from the perspective of a single agent since the other agents also change their policies during training. To analyze the non-stationarity in decentralized policy optimization, we first consider the Markov chain for the local state $s^k$ induced by the underlying MDP for agent $k$. When all agents' policies are updated from $\pi_1, ..., \pi_N$ to $\tilde{\pi}_1, ..., \tilde{\pi}_N$, the state transition distribution of this Markov chain also shifts. We denote such transition shift from $s^k$ to $\tilde{s}^k$ for agent $k$ as

$$\Delta_{\pi_1,...,\pi_N}^{\tilde{\pi}_1,...,\tilde{\pi}_N}(\tilde{s}^k|s^k) \triangleq \sum_{a^k}\left[p_{\tilde{\pi}_1,...,\tilde{\pi}_N}(\tilde{s}^k|s^k, a^k)\tilde{\pi}_k(a^k|s^k) - p_{\pi_1,...,\pi_N}(\tilde{s}^k|s^k, a^k)\pi_k(a^k|s^k)\right],$$

where $p_{\pi_1,\ldots,\pi_N}$ and $p_{\tilde{\pi}_1,\ldots,\tilde{\pi}_N}$ refer to the transition dynamics before and update $\pi$ is updated. The state transition shift consists of two parts: an *exogenous part*, which is caused by the update of other agents' policies (i.e., the change of transition dynamics from $p_{\pi_k}$ to $p_{\tilde{\pi}_k}$), and an *endogenous part*, which is contributed by the update of the agent's own policy (i.e., the change of agent $k$'s policy from $\pi_k$ to $\tilde{\pi}_k$). See the appendix A.2 for detailed analysis. The exogenous shift breaks the assumption in the monotonic improvement guarantee (Schulman et al., 2015) that the MDP is stationary, i.e., the state transition shift arises only endogenously from the agent's policy changes. As a result, Theorem 1 no longer holds if one optimizes with independent ratios as in IPPO and MAPPO.

### 4.3 MONOTONIC POLICY IMPROVEMENT FOR INDEPENDENT RATIOS

We now provide a new analysis for optimization with independent ratios. As the above analysis suggests that the exogenous transition shift breaks the trust region analysis in TRPO, we instead consider how to handle this exogenous shift in training. Specifically, since the exogenous shift is caused by the changes of other agents' policies, it can be controlled by constraining the update of other agents' policies in centralized training.

**Proposition 1.** *In a Dec-MDP, the transition shift* $\Delta_{\pi_1,\ldots,\pi_N}^{\tilde{\pi}_1,\ldots,\tilde{\pi}_N}(\tilde{s}^k|s^k)$ *decomposes as follows:*

$$\Delta_{\pi_1,\ldots,\pi_N}^{\tilde{\pi}_1,\ldots,\tilde{\pi}_N}(\tilde{s}^k|s^k) = \Delta_{\pi_1,\pi_2,\ldots,\pi_N}^{\tilde{\pi}_1,\pi_2,\ldots,\pi_N}(\tilde{s}^k|s^k) + \Delta_{\tilde{\pi}_1,\pi_2,\pi_3,\ldots,\pi_N}^{\tilde{\pi}_1,\tilde{\pi}_2,\pi_3,\ldots,\pi_N}(\tilde{s}^k|s^k) + \ldots + \Delta_{\tilde{\pi}_1,\ldots,\tilde{\pi}_{N-1},\pi_N}^{\tilde{\pi}_1,\ldots,\tilde{\pi}_{N-1},\tilde{\pi}_N}(\tilde{s}^k|s^k).$$

The proof is given in the appendix A.3.1. This proposition implies that the state transition shift at local observation $s^k$ is caused by the shifts arising from all decentralized policies. This decomposition inspires the derivation of a new monotonic improvement guarantee for decentralized policy optimization by enforcing the trust region over joint policies.

Define the surrogate objective for decentralized policy $\pi_k$ as

$$L_{\pi_1,\pi_2,\ldots,\pi_N}^{(k)}(\tilde{\pi}_{k'}) = \sum_{s^k} \rho_{\pi_1,\pi_2,\ldots,\pi_N}(s^k) \sum_{a^k} \tilde{\pi}_{k'}(a^k|s^k) A_{\pi_{k'}}(s^k, a^k),$$

and the expected return of decentralized policy $\pi_k$ as

$$\eta_{\pi_1,\ldots,\pi_N}(\pi_k) = \mathbb{E}_{(s^k,a^k)\sim\rho_{\pi_1,\ldots,\pi_N}(s^k,a^k)}\left[r_k(s^k,a^k)\right].$$

In practice, $r_k(s^k, a^k)$ is usually unknown. As the state transition shift decomposes into the sum of transition shifts caused by each decentralized policy, we can bound this state transition shift with a centralized trust region constraint as in equation 5.

**Theorem 2.** *Let* $\alpha = D_{\mathrm{TV}}^{\max}(\boldsymbol{\pi}, \tilde{\boldsymbol{\pi}})$. *Then the following bound holds :*

$$\eta_{\tilde{\pi}_1,\ldots,\tilde{\pi}_N}(\tilde{\pi}_k) \geq \eta_{\pi_1,\ldots,\pi_N}(\pi_k) + \sum_{k'=1}^{N} L_{\pi_1,\ldots,\pi_N}^{(k)}(\tilde{\pi}_{k'}) - \frac{4\epsilon\gamma}{(1-\gamma)^2}\alpha^2 \quad \forall k,$$

*where* $\epsilon = \max_{k \in \mathcal{N}} \max_{s^k,a^k} |A_{\pi_k}(s^k, a^k)|$.

The proof is given in the appendix A.3.2. This theorem implies that, for sufficiently small $\alpha$, the performance increase of a decentralized policy $\pi_k$ is lower bounded by the sum of surrogate objectives for each decentralized policy with respect to the samples generated by $\pi_k$. In other words, if the trust region is enforced, the sum of surrogate objectives yields an approximate lower bound for $\eta_{\tilde{\pi}_1,\tilde{\pi}_2,\ldots,\tilde{\pi}_N}(\tilde{\pi}_k)$, which holds for any decentralized policy $\tilde{\pi}_k$.

Theorem 2 differs from Theorem 1 in three respects. First, the lower bound for one decentralized policy effectively relies on surrogate objectives for all agents, since the update of one agent's policy affects all other agents' transition probability. Therefore, to improve the performance for policy $\pi_k$, we can simultaneously maximize $L_{\pi_1,\ldots,\pi_N}^{(k)}(\tilde{\pi}_1) + L_{\pi_1,\ldots,\pi_N}^{(k)}(\tilde{\pi}_2) + \ldots + L_{\pi_1,\ldots,\pi_N}^{(k)}(\tilde{\pi}_N)$. Second, unlike the surrogate objective in Theorem 1, the new surrogate objective implicitly contains an independent ratio $\lambda_{\tilde{\pi}_k} \triangleq \frac{\tilde{\pi}_k(a^k|s^k)}{\pi_k(a^k|s^k)}$ as it can be rewritten as follows: $L_{\pi_1,\pi_2,\ldots,\pi_N}^{(k)}(\tilde{\pi}_{k'}) = \mathbb{E}_{(s^k,a^k)}\left[\frac{\tilde{\pi}_{k'}(a^k|s^k)}{\pi_{k'}(a^k|s^k)}A_{\pi_k}(s^k,a^k)\right]$. Third, the additional term $\frac{4\epsilon\gamma}{(1-\gamma)^2}\alpha^2$ requires computing the total variation between joint policies $\boldsymbol{\pi}(\boldsymbol{a}|\boldsymbol{s})$ and $\tilde{\boldsymbol{\pi}}(\boldsymbol{a}|\boldsymbol{s})$, rather than the policies that are directly optimized. We show in the next section that, in centralized training, this requirement is easily satisfied as the magnitude of the constraint on the update is proportional to the number of agents.

## 5 REALIZING TRUST REGIONS VIA BOUNDING INDEPENDENT RATIOS

Theorem 2 indicates that the centralized trust region is crucial to guarantee monotonic improvement. In this section, we show that bounding independent ratios is an effective way to enforce such a centralized trust region constraint, and this enforcement requires taking into account the number of agents. To achieve this, we first present two lemmas about $D_{\text{TV}}$ divergence.

**Proposition 2.** *In a Dec-MDP, $D_{\text{TV}}^{\max}(\boldsymbol{\pi}, \tilde{\boldsymbol{\pi}}) \leq \sum_{k=1}^{N} D_{\text{TV}}^{\max}(\pi_k, \tilde{\pi}_k)$.*

This proposition is a direct result of the fact that the joint policy in a Dec-MDP factors as a product of decentralized polices, i.e., $\boldsymbol{\pi} = \prod_{k=1}^{N} \pi_k$ .

**Proposition 3.** $D_{\text{TV}}^{\max}(\pi_k, \tilde{\pi}_k) = \max_{s \in \mathcal{S}} \sum_{\tilde{\pi}_k(a^k|s^k) \geq \pi_k(a^k|s^k)} \left[ \tilde{\pi}_k(a^k|s^k) - \pi_k(a^k|s^k) \right]$.

This useful identity follows from a property of $D_{\text{TV}}$: $D_{\text{TV}}(\mu(x), \nu(x)) = \sum_{\mu(x) > \nu(x)} [\mu(x) - \nu(x)]$ where $\mu$ and $\nu$ are two distributions. This proposition indicates that a decentralized trust region is also defined by the sum of probability differences over a special subset. We use this to upper bound the trust region in the following analysis.

**Assumption 1.** *Assume the advantage function $A_{\pi_k}(s_k, a_k)$ converges to a fixed point for $\forall k$.*

**Theorem 3.** *If independent ratios $\lambda_k \triangleq \frac{\tilde{\pi}_k(a^k|s^k)}{\pi_k(a^k|s^k)}$ are within the range $[1 - \epsilon_k, 1 + \epsilon_k]$ for $\forall k \in \mathcal{N}$, then the following bound holds: $D_{\text{TV}}^{\max}(\pi_k(\cdot|s), \tilde{\pi}_k(\cdot|s)) < \epsilon_k$; $D_{\text{TV}}^{\max}(\boldsymbol{\pi}(\cdot|\boldsymbol{s}), \tilde{\boldsymbol{\pi}}(\cdot|\boldsymbol{s})) < \sum_{k=1}^{N} \epsilon_k$.*

This theorem comes from that fact that optimizing $\tilde{\pi}(s, a)$ with respect to a converged $A(s, a)$ leads to $\tilde{\pi}(a|s) > \pi(a|s)$ if $A(s, a) > 0$, $\forall s, a$ in actor-critic algorithms (Konda & Tsitsiklis, 2000). Thus, based on Proposition 3, we have the following

$$D_{\text{TV}}^{\max}(\pi_k, \tilde{\pi}_k) = \max_{s^k} \sum_{\substack{a^k \in \mathcal{A}^k \\ A_k(s^k, a^k) > 0}} [\tilde{\pi}_k(a^k|s^k) - \pi_k(a^k|s^k)] \leq \max_{s^k} \sum_{\substack{a \in \mathcal{A}^k \\ A_k(s^k, a^k) > 0}} [\epsilon_k \pi_k(a^k|s^k)] < \epsilon_k.$$

The equation is from Proposition 3 by considering $A(s, a) > 0$ such that $\tilde{\pi}(a|s) > \pi(a|s)$. The first inequality is a result of bounded ratios and the second is from $\sum_{a \in \{a : A(s,a) > 0\}} [\pi(a|s)] < 1$ (considering the lower bound yields the same analysis.) Furthermore, the trust region constraint over joint policies is a direct result of Proposition 2. The detailed proof is given in the appendix A.3.3. As $D_{\text{TV}}$ is a bounded divergence between $[0, 1]$, the ratio guarantee makes sense when $\epsilon_k \leq 1.0$.

Theorem 3 implies that bounding independent ratios $\frac{\tilde{\pi}_k(a^k|s^k)}{\pi_k(a^k|s^k)}$ with $[1 - \epsilon_k, 1 + \epsilon_k]$ amounts to enforcing a trust region constraint with size $\epsilon$ over decentralized policies. Thus, to enforce the centralized trust region constraint over joint policies, i.e., $D_{\text{TV}}^{\max}(\boldsymbol{\pi}, \tilde{\boldsymbol{\pi}}) \leq \alpha$ as in Theorem 2, one can consider bounding independent ratios according to the number of agents, e.g., $\lambda_k \triangleq \frac{\tilde{\pi}_k(a^k|s^k)}{\pi_k(a^k|s^k)} \in [1 - \frac{\alpha}{N}, 1 + \frac{\alpha}{N}]$. In the next section, we present practical implementations of these ratio constraints.

## 6 INSTANTIATING RATIO CONSTRAINTS

In this section, we show that IPPO and MAPPO effectively satisfy the conditions of Theorem 2 and Theorem 3. Specifically, the independent clipping and parameter sharing used by IPPO and MAPPO are useful ways to approximate the ratio constraints in Theorem 3 and to optimize the surrogate objective in Theorem 2. Furthermore, we show that the surrogate objectives optimized in IPPO and MAPPO are essentially equivalent when their critics converge to a fixed point.

### 6.1 OPTIMIZING SURROGATE OBJECTIVES

The objective is to update all agents' policies simultaneously with the experience from all agents, which can be further simplified with parameter sharing (Gupta et al., 2017):

$$\max_{\theta} \quad \sum_k \sum_{s^k} \rho_{\pi^1, \pi^2, \ldots, \pi^N}(s^k) \sum_{u^k} \tilde{\pi}_\theta(u^k|s^k) A_\phi(s^k, u^k), \tag{7}$$

$$\text{s.t.} \quad D_{TV}^{\max}(\boldsymbol{\pi}(\cdot|\boldsymbol{s}), \tilde{\boldsymbol{\pi}}(\cdot|\boldsymbol{s})) \leq \alpha, \tag{8}$$

where $\theta$ and $\phi$ are the shared parameters for decentralized policies and critics. Furthermore, to effectively optimize the surrogate objective, we can clip the probability ratios of each decentralized policies to form a lower bound of the objective in equation 7, similar to PPO (Schulman et al., 2017). Namely, with independent ratios $\lambda^k = \frac{\tilde{\pi}_\theta(u^k|s^k)}{\pi_\theta(u^k|s^k)}$ $\forall k \in \mathcal{N}$, we can optimize the following objective:

$$\max_\theta \quad \sum_k \mathbb{E}_{(s^k,u^k)\sim\rho(s^k,u^k)}\big[\min\big(\lambda^k A^k, \mathrm{clip}(\lambda^k, 1-\epsilon, 1+\epsilon)A^k\big)\big],$$

which is exactly the objective used by IPPO and MAPPO.

## 6.2 ENFORCING THE TRUST REGION CONSTRAINT

Proposition 2 implies that, in centralized training, one way to enforce trust region constraint is to delegate the centralized trust region constraint to each agent, such that the update of each decentralized policy $\pi_k(a^k|s^k)$ is bounded. Therefore, to enforce the centralized trust region constraint, one can impose a sufficient condition as follows,

$$D_{\mathrm{TV}}^{\max}\big(\pi_\theta(\cdot|s^k), \tilde{\pi}_\theta(\cdot|s^k)\big) \leq \frac{\alpha}{N} \quad \forall k \in \mathcal{N}, \tag{9}$$

which suggests that enforcement of the centralized trust region constraint translates to a decentralized trust region constraint if the trust region is specified properly according to the number of agents. Furthermore, based on Theorem 3, bounding independent ratios is an effective way to enforce the trust region constraint. One can thus impose a sufficient condition to constrain independent ratios $\lambda^k$ such that $\lambda^k \in [1 - \frac{\alpha}{N}, 1 + \frac{\alpha}{N}]$, where $N$ is the number of agents in training. Clipping is one of many ways to achieve this sufficient condition but itself is a heuristic approximation so often fails to bound ratios exactly within the ranges. In practice, one would need to tune the the clipping range and the number of epochs so the ratios can be properly bounded. We show in the experiment section that good values of the hyperparameters for the clipping range are highly sensitive to the number of agents, as these hyperparameters, together with the number of agents, effectively determine the size of the centralized trust region.

## 6.3 LEARNING ADVANTAGE FUNCTIONS

We now look at the training of the advantage function, where IPPO and MAPPO differ. IPPO trains a decentralized advantage function, while MAPPO trains a centralized one that incorporates centralized state information. Assume a stationary distribution of $(s^k, a^k)$ exists. From Lyu et al. (2021), we have the following:

**Proposition 4.** *(Lemma 1 & 2 in Lyu et al. (2021)) Training of centralized critic and $k$-th decentralized critic admits unique fixed points $Q^\pi(s^k, s^{-k}, a^k, a^{-k})$ and $\mathbb{E}_{s^{-k},a^{-k}}[Q^\pi(s^k, s^{-k}, a^k, a^{-k})]$ respectively, where $Q^\pi$ is the true expected return under the joint policy $\pi$.*

Accordingly, based on the definition, the centralized value function is $V(\boldsymbol{s}) = V(s^k, s^{-k}) = \mathbb{E}_{a^k,a^{-k}}[Q^\pi(s^k, s^{-k}, a^k, a^{-k})]$ and the decentralized one is $V(s^k) = \mathbb{E}_{s^{-k},a^k,a^{-k}}[Q^\pi(s^k, s^{-k}, u^k, a^{-k})] = \mathbb{E}_{s^{-k}}[V(s^k, s^{-k})] = \mathbb{E}_{s^{-k}}[V(\boldsymbol{s})]$. Thus, we have $A^{\mathrm{IPPO}}(s^k, a^k) = A^{\mathrm{MAPPO}}(s^k, a^k)$ (and so IPPO and MAPPO objectives are equivalent) given that the underlying critics converge to a fixed point.

## 7 EXPERIMENTS

We consider the StarCraft Multi-Agent Challenge (SMAC) (Samvelyan et al., 2019) for our empirical analysis as it provides a wide range of multi-agent tasks with varied difficulty and numbers of agents. We first show that clipping is an effective way to constraint ratios when the number of optimization epochs and the learning rate are properly specified. Furthermore, we show that clipping also requires taking into account the number of agents such that the centralized trust region can be properly enforced. We then empirically demonstrate that bounding independent ratios in effect enforces the trust region over joint policies. Finally, we present results showing that IPPO and MAPPO perform equivalently on SMAC maps with varied difficulty and numbers of agents.

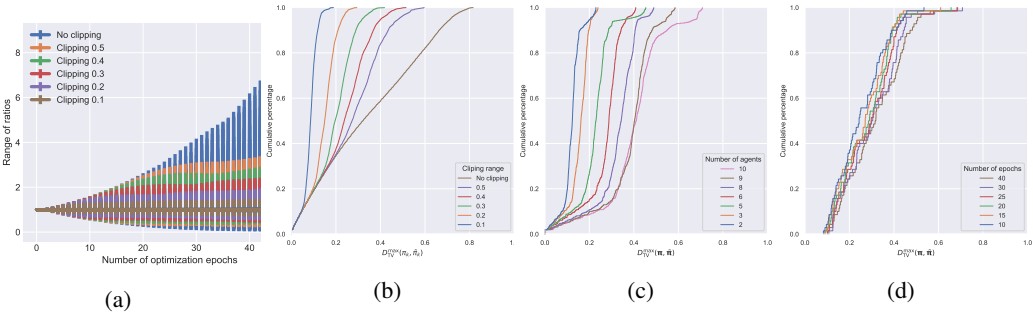

(a)          (b)          (c)          (d)

Figure 1: Trust region with respect to the clipping range, the number of agents and the number of optimization epochs: (a) Ratio ranges for 5 agents with the number of optimization epochs; (b) Cumulative percentage of decentralized trust region as the clipping value varies; (c) Cumulative percentage of centralized trust region as the number of agents varies (clipping at $0.1$); and (d) Cumulative percentage of centralized trust region with optimization epochs (clipping at $0.1$).

**Clipping and ratio ranges.** Theorem 3 indicates that bounding the independent ratios amounts to enforcing a trust region constraint over joint policies. We empirically show that independent ratio clipping approximately bounds the independent ratios in the training if some hyperparameters are properly set. We train decentralized policies on one map, *2s3z*, and clip the independent ratios when optimizing the surrogate objective. Figure 1a shows how the the max and min of the ratios changes according to the number of optimization epochs with different clipping values. Independent ratio clipping can effectively constrain the range of ratios only when the number of optimization epochs and the clipping range are properly specified. In particular, the range of independent ratios grows as the number of optimization epochs increases. This growth is slower when the clipping range is smaller, e.g., $\epsilon = 0.1$. Furthermore, the clipping range may not strictly bound ratios between $[1 - \epsilon, 1 + \epsilon]$: when the clipping range is $0.1$, the independent ratios can exceed $1.2$; and the independent ratios can even grow up to $1.6$ when the clipping range is $0.3$.

**Ratio clipping and trust region constraint.** Next, we show that the trust region defined by the total variation is empirically bounded by independent ratio clipping, and this bound is also proportional to the number of agents. Specifically, we compute the maximum total variation divergence $D_{\mathrm{TV}}^{\max}$ over empirical samples collected by the behavior policy during the first round of actor update, which contains 100 optimization epochs, and report the distribution of $D_{\mathrm{TV}}^{\max}$. Figure 1b shows the distribution of $D_{\mathrm{TV}}^{\max}$ over decentralized policies when clipping range varies. For clipping at $0.1$, all $D_{\mathrm{TV}}^{\max}$ values are smaller than $0.2$, meaning that the trust region is effectively enforced to be small. As the clipping range increases, more $D_{\mathrm{TV}}^{\max}$ values exceed $0.3$. For the case without clipping, $D_{\mathrm{TV}}^{\max}$ almost uniformly distributes among $[0.0, 0.8]$, implying trust region is no longer enforced. Figure 1c presents the distribution of $D_{\mathrm{TV}}^{\max}$ over joint polices for clipping at $0.1$, on maps with different number of agents. See appendix Table 1 for more details. The $D_{\mathrm{TV}}^{\max}(\boldsymbol{\pi}, \tilde{\boldsymbol{\pi}})$ is estimated by summing up the empirical total variation distances $D_{\mathrm{TV}}^{\max}(\pi_k, \tilde{\pi}_k)$ over all agents. The $D_{\mathrm{TV}}^{\max}(\boldsymbol{\pi}, \tilde{\boldsymbol{\pi}})$ grows almost proportionally with the number of agents, indicating that enforcing the centralized trust region with independent ratios clipping also requires considering the number of agents. Figure 1d presents the distribution of $D_{\mathrm{TV}}^{\max}$ over joint polices with different numbers of epochs for clipping at $0.1$. Compared to the number of agents, the number of epochs has less impact on the trust region.

**Independent ratios clipping on SMAC** Figure 2 shows the empirical returns and trust region estimates with different ratio clipping values across different maps in SMAC. [1] Notably, when the clipping value is small, e.g., $\epsilon = 0.1$, the joint total variation distance, i.e., the centralized trust region, can be effectively bounded, as in the second row in Figure 2. The empirical returns corresponding to $\epsilon = 0.1$ are thus improved monotonically, outperforming all other returns consistently in four maps. Moreover, as the number of agents increases, the trust region enforced by clipping value $\epsilon = 0.1$ in the initial training phase also grows from less than $0.3$ to more than $0.5$. In contrast, for clipping at

---

[1] Trained via decentralized advantage, i.e., IPPO. Results with centralized advantage are similar, as presented in the appendix A.4. Unlike Yu et al. (2021), the value function is not clipped in our experiments.

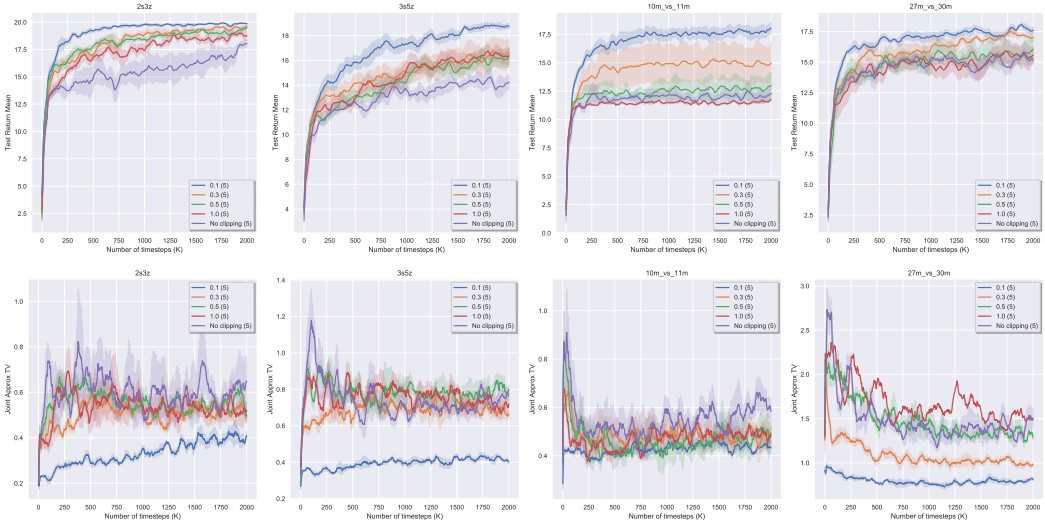

Figure 2: Empirical returns and trust region estimates for independent ratios clipping.

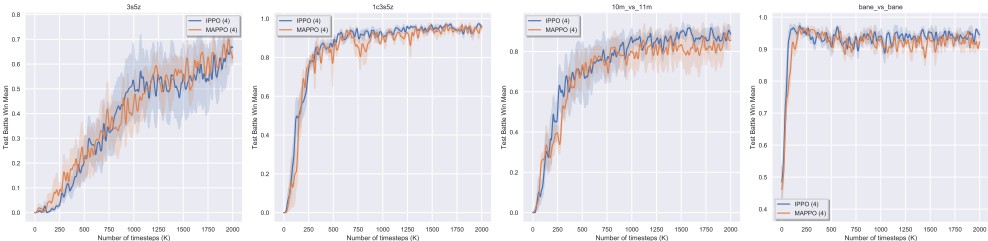

Figure 3: Contrasting IPPO and MAPPO across different maps.

0.5 and 1.0, the learning quickly plateaus at local optima, especially on maps with many agents, e.g., 10m_vs_11m and 27m_vs_30m, which shows that the policy performance $\eta(\pi_k)$ is closely related to the enforcement of trust region. We also apply the same clipping values to joint clipping and independent clipping, see Appendix A.6 for more analysis.

**IPPO and MAPPO**  We show that the empirical performance of IPPO and MAPPO are very similar despite the fact that the advantage functions are learned differently. We evaluate IPPO and MAPPO on maps of varied difficulty and numbers of agents. We heuristically set the clipping range based on the number of agents. Specifically, we set the clipping range $\epsilon$ for 3s5z, 1c3s5z, 10m_vs_11m, and bane_vs_bane, as 0.1, 0.1, 0.1, and 0.05, respectively. The results are presented in Figure 3. On the four maps considered, IPPO and MAPPO perform similarly. This phenomenon can be observed in Yu et al. (2021), which provides more empirical comparisons between IPPO and MAPPO. Such comparable performance also implies that, for actor-critic methods in MARL, the way of training critics could be less crucial than enforcing the trust region constraint.

## 8  CONCLUSION

In this paper, we present a new monotonic improvement guarantee for optimizing decentralized policies in cooperative MARL. We show that, despite the non-stationarity in IPPO and MAPPO, a monotonic improvement guarantee still arises from enforcing the trust region constraint over joint policies. This guarantee provides a theoretical understanding of the strong performance of IPPO and MAPPO. Furthermore, we provide a theoretical foundation for proximal ratio clipping by showing that a trust region constraint can be effectively enforced in a principled way by bounding independent ratios based on the number of agents in training. Finally, our empirical results support the hypothesis

that the strong performance of IPPO and MAPPO is a direct result of enforcing such a trust region constraint via clipping in centralized training.

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

# A APPENDIX

## A.1 JOINT RATIO PPO

---
**Algorithm 1** Joint Ratio PPO (JR-PPO)

---
**for** iteration $i = 0, 1, 2, \ldots$ **do**

    Roll out decentralized policies $[\pi_1, \pi_2, ..., \pi_N]$ in environment;

    Compute centralized advantage estimates $A(\boldsymbol{s}, \boldsymbol{a})$;

    Compute joint ratios $\boldsymbol{\lambda}_{\tilde{\pi}} = \frac{\tilde{\boldsymbol{\pi}}(\boldsymbol{a}|\boldsymbol{s})}{\boldsymbol{\pi}(\boldsymbol{a}|\boldsymbol{s})} = \prod_{k=1}^{N}\left[\frac{\tilde{\pi}_k(a^k|s^k)}{\pi_k(a^k|s^k)}\right]$;

    Optimize the surrogate objective $\max_{\tilde{\boldsymbol{\pi}}} \mathbb{E}\big[\min\big(\boldsymbol{\lambda}_{\tilde{\pi}}A(\boldsymbol{s}, \boldsymbol{a}), \text{clip}(\boldsymbol{\lambda}_{\tilde{\pi}}, 1 - \epsilon, 1 + \epsilon)A(\boldsymbol{s}, \boldsymbol{a})\big)\big]$.

**end for**

---

## A.2 STATIONARITY ASSUMPTION IN TRPO

The single-agent TRPO relies on the following analysis:

$$
\begin{aligned}
L_\pi(\tilde{\pi}) - L_\pi(\pi) &= \sum_s \rho(s) \sum_a \big(\tilde{\pi}(a|s) - \pi(a|s)\big) A_\pi(s, a) \\
&= \sum_s \rho(s) \sum_a \big(\tilde{\pi}(a|s) - \pi(a|s)\big)\big[r(s) + \sum_{s'} p(s'|s, a)\gamma v(s') - v(s)\big] \\
&= \sum_s \rho(s) \sum_{s'} \sum_a \big(\tilde{\pi}(a|s) - \pi(a|s)\big)p(s'|s, a)\gamma v(s') \\
&= \sum_s \rho(s) \sum_{s'} \sum_a \big(\tilde{\pi}(a|s)p(s'|s, a) - \pi(a|s)p(s'|s, a)\big)\gamma v(s') \\
&= \sum_s \rho(s) \sum_{s'} \big(p_{\tilde{\pi}}(s'|s) - p_\pi(s'|s)\big)\gamma v(s').
\end{aligned}
$$

This analysis is based on the assumption that $p(s'|s, a)$ remains the same before and after $\pi$ is updated, such that transition shift $p_{\tilde{\pi}}(s'|s) - p_\pi(s'|s)$ is only caused by the agent's policy update, i.e., endogenously. Such analysis no longer holds when the transition dynamics $p(s'|s, a)$ are non-stationary: $p_{\tilde{\pi}}(s'|s, a) \neq p_\pi(s'|s, a)$.

## A.3 PROOFS

### A.3.1 PROOF OF PROPOSITION 1

*Proof.* Assume agent $k$'s policy $\pi^k$ is executed independently of other agents policies $\pi^{-k}$, we have

$$
\begin{aligned}
&\Delta_{\pi^1, \ldots, \pi^N}^{\tilde{\pi}^1, \ldots, \tilde{\pi}^N}(\tilde{s}^k|s^k) \\
&= \sum_{\substack{\tilde{s}^{-k}, s^{-k} \\ a^k, a^{-k}}} p(\tilde{s}^k, \tilde{s}^{-k}|s^k, s^{-k}, a^k, a^{-k})\big[\tilde{\pi}^k(a^k|s^k)\tilde{\pi}^{-k}(a^{-k}|s^{-k}) - \pi^k(a^k|s^k)\pi^{-k}(a^{-k}|s^{-k})\big] \\
&= \sum_{\substack{\tilde{s}^{-k}, s^{-k} \\ a^k, a^{-k}}} p(\tilde{s}^k, \tilde{s}^{-k}|s^k, s^{-k}, a^k, a^{-k}) \cdot \big[\underbrace{\tilde{\pi}^k(a^k|s^k)\tilde{\pi}^{-k}(a^{-k}|s^{-k}) - \tilde{\pi}^k(a^k|s^k)\pi^{-k}(a^{-k}|s^{-k})}_{\text{exogenous}} \\
&\quad + \underbrace{\tilde{\pi}^k(a^k|z^k)\pi^{-k}(a^{-k}|s^{-k}) - \pi^k(a^k|s^k)\pi^{-k}(a^{-k}|s^{-k})}_{\text{endogenous}}\big].
\end{aligned}
$$

The above decomposition can be repeated such that the exogenous part can be translated into endogenous parts that are specific to each agent. Specifically, repeat the decomposition for the

exogenous part by considering agent $k'$ ($k' \neq k$):

$$\tilde{\pi}^k(a^k|s^k)\tilde{\pi}^{-k}(a^{-k}|s^{-k}) - \tilde{\pi}^k(a^k|s^k)\pi^{-k}(a^{-k}|s^{-k})$$

$$=\tilde{\pi}^k(a^k|s^k)\big[\tilde{\pi}^{k'}(a^{k'}|s^{k'})\tilde{\pi}^{-\{k,k'\}}(a^{-\{k,k'\}}|s^{-\{k,k'\}}) - \pi^{k'}(a^{k'}|s^{k'})\pi^{-\{k,k'\}}(a^{-\{k,k'\}}|s^{-\{k,k'\}})\big]$$

$$=\tilde{\pi}^k(a^k|s^k)\big[\underbrace{\tilde{\pi}^{k'}(a^{k'}|s^{k'})\tilde{\pi}^{-\{k,k'\}}(a^{-\{k,k'\}}|s^{-\{k,k'\}}) - \tilde{\pi}^{k'}(a^{k'}|s^{k'})\pi^{-\{k,k'\}}(a^{-\{k,k'\}}|s^{-\{k,k'\}})}_{\pi^k\text{-exogenous}}$$

$$+ \underbrace{\tilde{\pi}^{k'}(a^{k'}|s^{k'})\pi^{-\{k,k'\}}(a^{-\{k,k'\}}|s^{-\{k,k'\}}) - \pi^{k'}(a^{k'}|s^{k'})\pi^{-\{k,k'\}}(a^{-\{k,k'\}}|s^{-\{k,k'\}})}_{\pi^k\text{-endogenous}}\big].$$

So on and so forth, one can decompose $\Delta^{\tilde{\pi}^1,...,\tilde{\pi}^N}_{\pi^1,...,\pi^N}(\tilde{s}^k|s^k)$ as follows:

$$\Delta^{\tilde{\pi}^1,...,\tilde{\pi}^N}_{\pi^1,...,\pi^N}(\tilde{s}^k|s^k) = \Delta^{\tilde{\pi}^1,\pi^2,...,\pi^N}_{\pi^1,\pi^2,...,\pi^N}(\tilde{s}^k|s^k) + \Delta^{\tilde{\pi}^1,\tilde{\pi}^2,\pi^3,...,\pi^N}_{\tilde{\pi}^1,\pi^2,\pi^3,...,\pi^N}(\tilde{s}^k|s^k) + ... + \Delta^{\tilde{\pi}^1,...,\tilde{\pi}^{N-1},\tilde{\pi}^N}_{\tilde{\pi}^1,...,\tilde{\pi}^{N-1},\pi^N}(\tilde{s}^k|s^k),$$

which implies that the state transition shift at local observation $s^k$ is caused by the shifts arising from all decentralized policies. $\square$

### A.3.2 PROOF OF THEOREM 2

*Proof.* This proof is based on the perturbation theory. Let $G^{s_i} = (1 + \gamma P^{s_i}_{\pi_1,\pi_2,...,\pi_N} + (\gamma P^{s_i}_{\pi_1,\pi_2,...,\pi_N})^2 + ... = (1 - \gamma P^{s_i}_{\pi_1,\pi_2,...,\pi_N})^{-1}$ and $\tilde{G}^{s_i} = (1 + \gamma P^{s_i}_{\tilde{\pi}_1,\tilde{\pi}_2,...,\tilde{\pi}_N} + (\gamma P^{s_i}_{\tilde{\pi}_1,\tilde{\pi}_2,...,\tilde{\pi}_N})^2 + ... = (1 - \gamma P^{s_i}_{\tilde{\pi}_1,\tilde{\pi}_2,...,\tilde{\pi}_N})^{-1}$ denote the distribution of state $s_i$ under $\pi_1, \pi_2, ..., \pi_N$ and $\tilde{\pi}_1, \tilde{\pi}_2, ..., \tilde{\pi}_N$. We will use the convention that $\rho$ (a density on state space) is a vector and $r$ (a reward function on state space) is a dual vector (i.e., linear functional on vectors), thus $r\rho$ is a scalar meaning the expected reward under density $\rho$. Note that $\eta(\pi) = rG\rho_0$, and $\eta(\tilde{\pi}) = r\tilde{G}\rho_0$. Denote the state shift of $s_i$ as $\Delta(s_i) = P^{s_i}_{\tilde{\pi}_1,\tilde{\pi}_2,...,\tilde{\pi}_N} - P^{s_i}_{\pi_1,\pi_2,...,\pi_N}$. Using the perturbation theory, we have the following

$$[G^{s_i}]^{-1} - [\tilde{G}^{s_i}]^{-1} = \gamma P^{s_i}_{\tilde{\pi}_1,\tilde{\pi}_2,...,\tilde{\pi}_N} - \gamma P^{s_i}_{\pi_1,\pi_2,...,\pi_N} = \gamma\Delta(s_i).$$

Left multiply by $G^{s_i}$ and right multiply by $\tilde{G}^{s_i}$:

$$\tilde{G}^{s_i} = G^{s_i} + \gamma G^{s_i}\Delta(s_i)\tilde{G}^{s_i}.$$

Substituting the right-hand side into $\tilde{G}^{s_i}$ gives

$$\tilde{G}^{s_i} - G^{s_i} = \gamma G^{s_i}\Delta(s_i)G^{s_i} + \gamma^2 G^{s_i}\Delta(s_i)G^{s_i}\Delta(s_i)\tilde{G}^{s_i}.$$

Consider decentralized policy $\pi_i$:

$$\eta_{\tilde{\pi}_1,\tilde{\pi}_2,...,\tilde{\pi}_N}(\tilde{\pi}_i) - \eta_{\pi_1,\pi_2,...,\pi_N}(\pi_i)$$
$$=r\tilde{G}^{s_i}\rho_0 - rG^{s_i}\rho_0 = r(\tilde{G}^{s_i} - G^{s_i})\rho_0$$
$$=\gamma rG^{s_i}\Delta(s_i)G^{s_i}\rho_0 + \gamma^2 rG^{s_i}\Delta(s_i)G^{s_i}\Delta(s_i)\tilde{G}^{s_i}\rho_0.$$

Let us first consider the leading term $\gamma rG^{s_i}\Delta(s_i)G^{s_i}\rho_0$,

$$\gamma rG^{s_i}\Delta(s_i)G^{s_i}\rho_0$$
$$=\sum_{s_i}\rho(s_i)\sum_{s_i'}(p_{\tilde{\pi}}(s_i'|s_i) - p_\pi(s_i'|s_i))\gamma v(s_i') = \sum_{s_i}\rho(s_i)\sum_{s_i'}\Delta^{\tilde{\pi}_1,...,\tilde{\pi}_N}_{\pi_1,...,\pi_N}(s_i'|s_i)\gamma v(s_i')$$
$$=\sum_{s_i}\rho(s_i)\sum_{s_i'}\Big[\Delta^{\tilde{\pi}_1,\pi_2,...,\pi_N}_{\pi_1,\pi_2,...,\pi_N}(s_i'|s_i) + \Delta^{\tilde{\pi}_1,\tilde{\pi}_2,\pi_3,...,\pi_N}_{\tilde{\pi}_1,\pi_2,\pi_3,...,\pi_N}(s_i'|s_i) + ... + \Delta^{\tilde{\pi}_1,...,\tilde{\pi}_{N-1},\tilde{\pi}_N}_{\tilde{\pi}_1,...,\tilde{\pi}_{N-1},\pi_N}(s_i'|s_i)\Big]\gamma v(s_i').$$

For one of these summation terms, we have the following [2]:

$$\sum_{s_i} \rho(s_i) \sum_{s_i'} \left[ \Delta_{\tilde{\pi}_1,\ldots,\tilde{\pi}_{j-1},\pi_j,\ldots,\pi_N}^{\tilde{\pi}_1,\ldots,\tilde{\pi}_{j-1},\tilde{\pi}_j,\ldots,\pi_N}(s_i'|s_i) \right] \gamma v(s_i') \tag{10}$$

$$= \sum_{s_i} \rho(s_i) \sum_{s_i'} \sum_{a_i} \left( p_{\tilde{\pi}_1,\ldots,\tilde{\pi}_{j-1},\pi_j,\ldots,\pi_N}(s_i'|s_i,a_i)\tilde{\pi}_j(a_i|s_i) - p_{\tilde{\pi}_1,\ldots,\tilde{\pi}_{j-1},\pi_j,\ldots,\pi_N}(s_i'|s_i,a_i)\pi_j(a_i|s_i) \right) \gamma v(s_i') \tag{11}$$

$$= \sum_{s_i} \rho(s_i) \sum_{a_i} \left( \tilde{\pi}_j(a_i|s_i) - \pi_j(a_i|s_i) \right) \sum_{s_i'} p_{\tilde{\pi}_1,\ldots,\tilde{\pi}_{j-1},\pi_j,\ldots,\pi_N}(s_i'|s_i,a_i)\gamma v(s_i') \tag{12}$$

$$= \sum_{s_i} \rho(s_i) \sum_{a_i} \left( \tilde{\pi}_j(a_i|s_i) - \pi_j(a_i|s_i) \right) \left[ r(s_i) + \sum_{s_i'} p_{\tilde{\pi}_1,\ldots,\tilde{\pi}_{j-1},\pi_j,\ldots,\pi_N}(s_i'|s_i,a_i)\gamma v(s_i') - v_{\substack{\tilde{\pi}_1,\ldots,\tilde{\pi}_{j-1},\\ \pi_j,\ldots,\pi_N}}(s_i) \right] \tag{13}$$

$$= \sum_{s_i} \rho(s_i) \sum_{a_i} \left( \tilde{\pi}_j(a_i|s_i) - \pi_j(a_i|s_i) \right) A_{\pi_j}(s_i,a_i) \tag{14}$$

$$= L_{\pi_1,\pi_2,\ldots,\pi_N}^{(i)}(\tilde{\pi}_j) - L_{\pi_1,\pi_2,\ldots,\pi_N}^{(i)}(\pi_j). \tag{15}$$

The derivation from line 13 to 14 is based on the following definition:

$$A_{\pi_j}(s_i,a_i) \triangleq r(s_i) + \sum_{s_i'} p_{\tilde{\pi}_1,\ldots,\tilde{\pi}_{j-1},\pi_j,\ldots,\pi_N}(s_i'|s_i,a_i)\gamma v(s_i') - v_{\substack{\tilde{\pi}_1,\ldots,\tilde{\pi}_{j-1},\\ \pi_j,\ldots,\pi_N}}(s_i),$$

where $v_{\substack{\tilde{\pi}_1,\ldots,\tilde{\pi}_{j-1},\\ \pi_j,\ldots,\pi_N}}(s_i) \triangleq r(s_i) + \gamma \sum_{s_i'} p_{\tilde{\pi}_1,\ldots,\tilde{\pi}_{j-1},\pi_j,\ldots,\pi_N}(s_i'|s_i,a_i) \sum_{a_i} \pi_j(a_i|s_i)v(s_i')$. which will result in $L_{\pi_1,\pi_2,\ldots,\pi_N}^{(i)}(\pi_j) = 0$ for $\forall j$. In fact, for line 13, $r(s_i)$ and $v_{\substack{\tilde{\pi}_1,\ldots,\tilde{\pi}_{j-1},\\ \pi_j,\ldots,\pi_N}}(s_i)$ can be interpreted as functions over $s_i$, which will be zero if integrated with $\sum_{a_i} \left( \tilde{\pi}_j(a_i|s_i) - \pi_j(a_i|s_i) \right)$. Note that the advantage function also conditions on the $\pi_j$.

Note that the advantage of $\pi_j$ with respect to $s_i, a_i$ in multi-agent RL is defined differently from the advantage function in the single agent case.

We can thus repeat the above decomposition iteratively and have the following:

$$\gamma r G^{s_i} \Delta(s_i) G^{s_i} \rho_0 = L_{\pi_1,\pi_2,\ldots,\pi_N}^{(i)}(\tilde{\pi}_1) - L_{\pi_1,\pi_2,\ldots,\pi_N}^{(i)}(\pi_1)$$
$$+ L_{\pi_1,\pi_2,\ldots,\pi_N}^{(i)}(\tilde{\pi}_2) - L_{\pi_1,\pi_2,\ldots,\pi_N}^{(i)}(\pi_2)$$
$$+ \ldots + L_{\pi_1,\pi_2,\ldots,\pi_N}^{(i)}(\tilde{\pi}_N) - L_{\pi_1,\pi_2,\ldots,\pi_N}^{(i)}(\pi_N).$$

Note that $L_{\pi_1,\pi_2,\ldots,\pi_N}^{(i)}(\pi_j) = 0$ for $\forall j$. Thus,

$$\gamma r G^{s_i} \Delta(s_i) G^{s_i} \rho_0 = L_{\pi_1,\pi_2,\ldots,\pi_N}^{(i)}(\tilde{\pi}_1) + L_{\pi_1,\pi_2,\ldots,\pi_N}^{(i)}(\tilde{\pi}_2) + \ldots + L_{\pi_1,\pi_2,\ldots,\pi_N}^{(i)}(\tilde{\pi}_N).$$

Next we bound the second term $\gamma^2 r G^{s_i} \Delta(s_i) G^{s_i} \Delta(s_i) \tilde{G}^{s_i} \rho_0$. First we consider the product $\gamma r G^{s_i} \Delta(s_i) = \gamma v \Delta(s_i)$. Consider the component of this dual vector:

$$|(\gamma r G^{s_i} \Delta(s_i)_s| = |\sum_{s_i'} (p_{\tilde{\pi}_i}(s_i'|s_i) - p_{\pi_i}(s_i'|s_i))\gamma v_i(s_i')|$$

$$= |\sum_{s_1',\ldots,s_N'} \sum_{s_1,\ldots,s_{i-1},s_{i+1},s_N} \sum_{a_1,\ldots,a_N} p(s_1',\ldots,s_N'|s_1,\ldots,s_N,a_1,\ldots,a_N)v_i(s_i') \left[ \prod_{j=1}^{N} \tilde{\pi}_j(a_i|s_i) - \prod_{j=1}^{N} \pi_j(a_i|s_i) \right]|$$

$$= |\sum_{s_1,\ldots,s_{i-1},s_{i+1},s_N} \sum_{a_1,\ldots,a_N} A_i(s_i,a_i) \left[ \prod_{j=1}^{N} \tilde{\pi}_j(a_i|s_i) - \prod_{j=1}^{N} \pi_j(a_i|s_i) \right]|.$$

Thus, we have

$$\|(\gamma r G^{s_i} \Delta(s_i)_s\| \le \sum_{s_1,\ldots,s_N} \sum_{a_1,\ldots,a_N} |A_i(s_i,a_i)| |\prod_{j=1}^{N} \tilde{\pi}_j(a_i|s_i) - \prod_{j=1}^{N} \pi_j(a_i|s_i)| \le 2\alpha\epsilon.$$

---

[2] Note that $A_{\pi_j}(s_i,a_i)$ in the analysis is defined with the transition dynamics $p_{\tilde{\pi}_1,\ldots,\tilde{\pi}_{j-1},\pi_j,\ldots,\pi_N}(s_i'|s_i,a_i)$.

We bound the other portion $G^{s_i}\Delta(s_i)\tilde{G}^{s_i}\rho_0$ using the $l_1$ operator norm:

$$\|G^{s_i}\Delta(s_i)\tilde{G}^{s_i}\rho_0\|_1 \le \frac{\|\Delta(s_i)\|_1}{(1-\gamma)^2}.$$

The $\|\Delta(s_i)\|_1$ can be bound as follows:

$$\|\Delta(s_i)\|_1 = \sum_{s_i',s_i}|p_{\tilde{\pi}_i}(s_i'|s_i) - p_{\pi_i}(s_i'|s_i)|$$

$$= \sum_{s_1',...,s_N'}\sum_{s_1,...,s_N}\sum_{a_1,...,a_N}p(s_1',...,s_N'|s_1,...,s_N,a_1,...,a_N)|\prod_{j=1}^N\tilde{\pi}_j(a_i|s_i) - \prod_{j=1}^N\pi_j(a_i|s_i)|$$

$$= \sum_{s_1,...,s_N}\sum_{a_1,...,a_N}|\prod_{j=1}^N\tilde{\pi}_j(a_i|s_i) - \prod_{j=1}^N\pi_j(a_i|s_i)| \le 2\alpha.$$

So we have that

$$\gamma^2|rG^{s_i}\Delta(s_i)G^{s_i}\Delta(s_i)\tilde{G}^{s_i}\rho_0| \le \gamma\|rG^{s_i}\Delta(s_i)\|_\infty\|G^{s_i}\Delta(s_i)\tilde{G}^{s_i}\rho_0\|_1 \le \frac{4\epsilon\gamma}{(1-\gamma)^2}\alpha^2.$$

$\square$

### A.3.3 PROOF OF THEOREM 3

*Proof.* Given the assumption that the advantage function $A(s,a)$ converges to a fixed point. we have the fact that optimizing $\tilde{\pi}(s,a)$ with respect to $A(s,a)$ leads to $\tilde{\pi}(a|s) > \pi(a|s)$ if $A(s,a) > 0$, $\forall s, a$. Consider independent ratios $\lambda_k \triangleq \frac{\tilde{\pi}_k(a^k|s^k)}{\pi_k(a^k|s^k)}$ which are within the range $[1-\epsilon_k, 1+\epsilon_k]$ for $\forall k \in \mathcal{N}$, Thus, based on Proposition 3, we have the following

$$D_{\text{TV}}^{\max}(\pi_k, \tilde{\pi}_k) = \max_{s^k}\sum_{\substack{a^k\in\mathcal{A}^k\\\tilde{\pi}(a|s)>\pi(a|s)}}[\tilde{\pi}_k(a^k|s^k) - \pi_k(a^k|s^k)]$$

$$\le \max_{s^k}\sum_{\substack{a^k\in\mathcal{A}^k\\A_k(s^k,a^k)>0}}[(1+\epsilon_k)\pi_k(a^k|s^k) - \pi_k(a^k|s^k)]$$

$$\le \max_{s^k}\sum_{\substack{a\in\mathcal{A}^k\\A_k(s^k,a^k)>0}}[\epsilon_k\pi_k(a^k|s^k)] < \epsilon_k.$$

The first inequality is a result of bounded ratios and the second is from $\sum_{a\in\{a:A(s,a)>0\}}[\pi(a|s)] < 1$. Furthermore, the trust region constraint over joint policies is a direct result of Proposition 2. One can also consider $A(s,a) < 0$, which leads to $\tilde{\pi}(a|s) < \pi(a|s)$ in optimization. Given that independent ratios are also lower bounded by $1-\epsilon$, the same conclusion can be reached for $D_{\text{TV}}^{\max}$. $\square$

### A.4 EXPERIMENT DETAILS AND MORE RESULTS

The number of agents in each is given in Table 1.

Test battle win mean of IPPO on maps with varied difficulty and numbers of agents is presented in Figure 4.

Empirical test battle win mean, test returns and trust region estimates of MAPPO on maps with varied difficult and numbers of agents are presented in Figure 5.

### A.5 ABLATIONS ON SMALL CLIPPING VALUES

We also present the ablation results for small clipping values, i.e., $< 0.1$, in Figure 6. It is true that a small clipping value results in a small trust region, and thus small clipping values, e.g., 0.08, 0.05 and 0.03, would be preferred for maps with a large number of agents, e.g., maps 10m_vs_11m (10

Table 1: Number of agents on maps.

| SMAC Map | Number of agents |
|----------|------------------|
| 2s_vs_1sc | 2 |
| 3s_vs_5z | 3 |
| 2s3z | 5 |
| 6h_vs_8z | 6 |
| 1c3s5z | 9 |
| 10m_vs_11m | 10 |

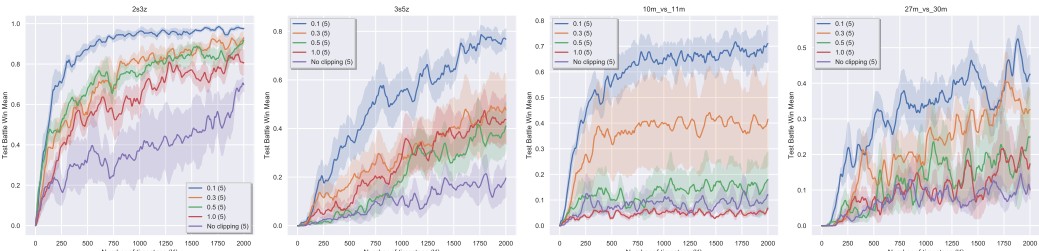

Figure 4: Test battle win mean of IPPO on maps with varied difficulty and numbers of agents

agents) and 27m_vs_30m (27 agents). However, when the clip value is too small, e.g., $\epsilon = 0.01$ in maps with 5 and 8 agents, the resultant trust region is also small and the update step in each iteration can thus be too small to improve the policy. Thus, one would need to trade off between the trust region constraint, to ensure monotonic improvement, and the policy update step, to ensure a sufficient parameter update at each iteration.

### A.6 COMPARISON BETWEEN JOINT RATIO CLIPPING AND INDEPENDENT RATIO CLIPPING

We apply the same clipping values to these two types of clipping, and use maps with many agents, i.e., 10m_vs_11m and 27m_vs_30m, to make the difference more salient (based on the theoretical results in the paper). The results are presented in Figure 7 and 8.

Compared to joint ratio clipping, the independent ratio clipping is more sensitive to the number of agents. In particular, for a small clipping value, e.g., $\epsilon = 0.1$, joint ratio clipping consistently produces better performance than independent ratio clipping, even when the number of agents changes from 10 to 27. As the clipping value increases to $0.5$, the performance gap between these two types of clipping becomes larger, which is also aligned with our theoretical analysis.

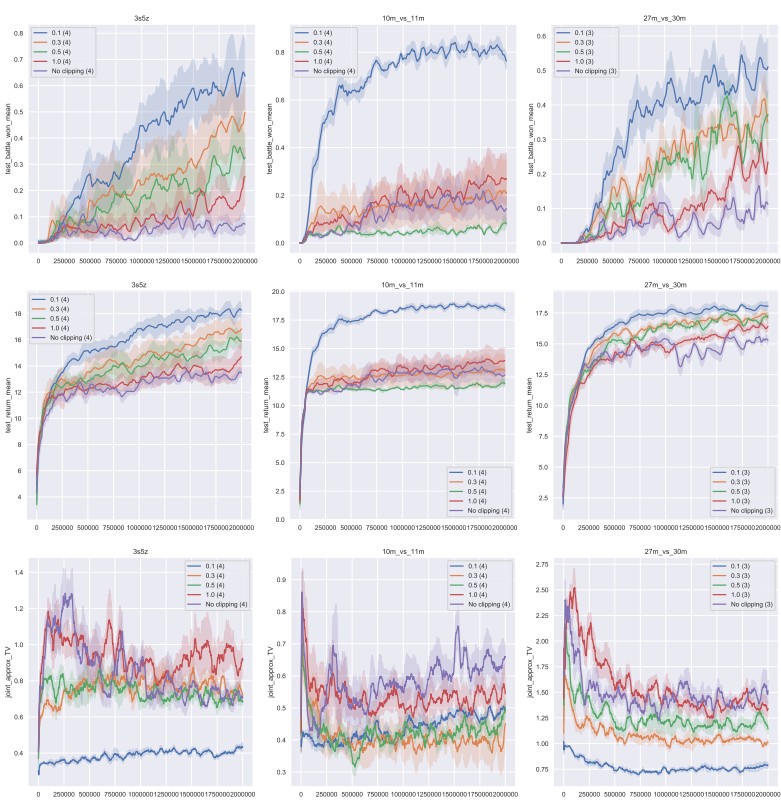

Figure 5: Empirical test battle win mean (first row), test returns (second row) and trust region estimates (third row) of MAPPO on maps with varied difficult and numbers of agents

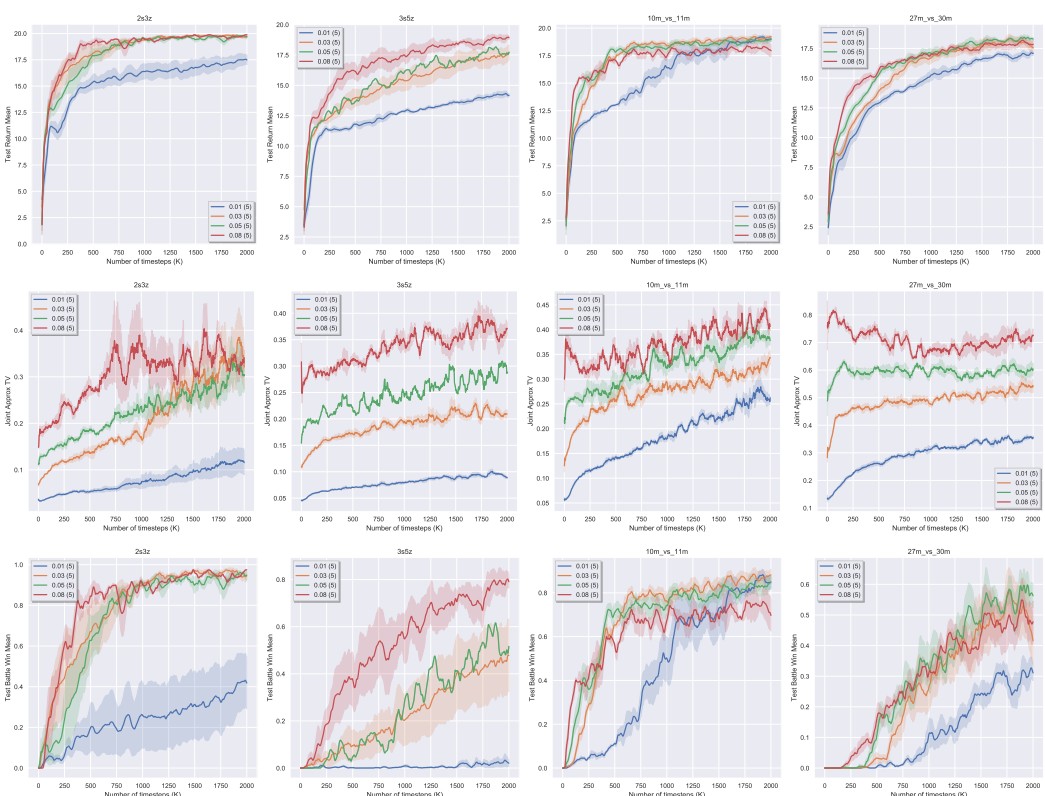

Figure 6: Empirical returns, trust region estimates and test battle win rate for small values of independent ratios clipping.

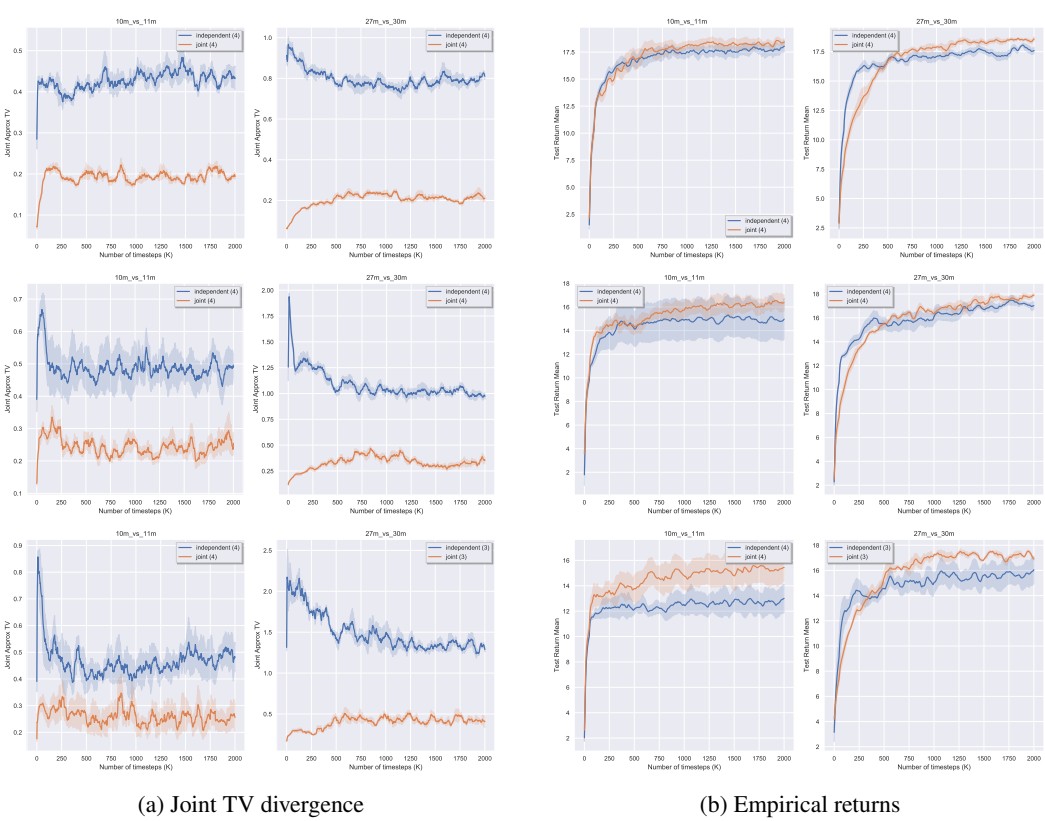

(a) Joint TV divergence

(b) Empirical returns

Figure 7: Joint divergence estimates and empirical returns for two types of ratio clipping at different clipping values: $0.1$ (first row), $0.3$ (first row) and $0.5$ (first row).

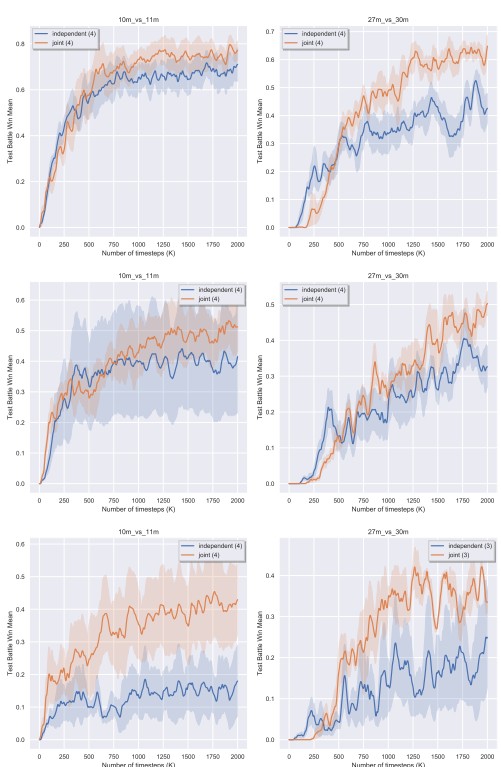

Figure 8: Test battle win rate for two types of ratio clipping at different clipping values: $0.1$ (first row), $0.3$ (first row) and $0.5$ (first row).

