# OpenReview forum: "Monotonic Improvement Guarantees under Non-stationarity for Decentralized PPO"
_ICLR.cc/2022/Conference — ICLR 2022 Submitted_

### Official Review · Reviewer_YJEa · 2021-11-02

**Correctness:** 1
**Technical Novelty And Significance:** 1
**Empirical Novelty And Significance:** 2
**Recommendation:** 3
**Confidence:** 4

**Main Review:**

**Strengths**

- Decentralized policy learning with monotonic joint policy improvement is a very important problem for MARL.
- A theoretical understanding for the good performance of IPPO and MAPPO is also desirable.

**Problems**

- The main contribution is Theorem 2, but I think there might be some mistakes in the proof of Theorem 2. In the deductions at the top of the page 14, from the fourth line to the fifth line in the equations,   $r(s_i) + \cdots$ **is not equal** to $A_{\pi_j}(s_i, a_i)$. Although you have mentioned that you use an approximation for $A_{\pi_j}(s_i,a_i)$ in the footnote, I think it is inappropriate to use such an approximation in the proof especially when this is the main contribution of this paper. Moreover, this approximation actually ignores the *non-stationarity* caused by independent learnings which is the major obstacle for this problem in theory.

This is the major issue of this paper. Please clarify if I misunderstood something.


**Summary Of The Paper:**

This paper tries to provide a theoretical monotonic improvement guarantees for  IPPO and MAPPO and shows enforcing independent trust region constraints could enforce the trust region constraint over joint policies. The empirical results are also provided to support the hypothesis.

**Summary Of The Review:**

The approximation is inappropriate which makes the contribution of the paper meaningless, unless I left something important.

### After rebuttal

There is an error in the proof of Theorem 2 which is claimed as the major contribution of this paper.  During rebuttal, the authors changed these equations (10 -15) several times, but unfortunately none of them are correct. In the latest revision (from equation 13 to 14), the $new$ definition of $A_{\pi_j}$ is given (the definition below equation 15), but it does not have any actual meaning (i.e., it is *not* an advantage function), just a function by definition. This DOES NOT accord with the learning of IPPO, MAPPO, or any other MARL methods I know.

Thus, I keep my score unchanged.

---

> ### Author Response · Authors · 2021-11-18
> **Thank you for the review!**
>
> Thank you for the review!
>
> We acknowledge that notations could be improved to avoid confusion and
> we will make them more explicit in the revised version.
> But there is no approximation to the advantage and,
> even more importantly, there is no error in the proof.
>
> Specifically, the advantage of $\pi_j$ with respect to $s_i, a_i$ in multi-agent RL is defined
> differently from the advantage function in the single agent case.
> We updated the manuscript to make it clearer (apologize that we cannot put the proof here as the latex formula cannot be parsed).
> Please refer to the supplementary materials for more details (in section REVIEWER YJEA).
> Thus, from the fourth line to the fifth line in the equations,
> the derivation is exact, and no approximation is applied.
> Intuitively, one can interpret this derivation as applying a series of single-agent perturbation analysis
> with a changing advantage function.
> This changing advantage function, however, does not affect the improvement guarantee as long as
> the maximum advantage, e.g., $\max_{k\in\mathcal{N}}\max_{s^k,a^k}\lvert A_{\pi_{k}}(s^k, a^k) \rvert$,
> can be bounded and a trust region constraint is enforced.
> This is the key idea to derive the monotonic improvement guarantee for MARL.
> In a nutshell, instead of fixating on the changing advantage function,
> we leverage the fact that the advantage itself should be bounded regardless,
> which then yields the monotonic improvement guarantee presented in the paper.

---

> > ### Comment · Reviewer_YJEa · 2021-11-22
> > **Further comments**
> >
> > Thanks for the reply. However, I am still not convinced by the response.
> >
> > Of course, you could define $A_{\pi_j}(s,a_i)$ like this in your proof (equation 13, in Appendix). But in this definition, the functions $v$ for the state $s^\prime$ and $s$ are different. Thus, the second term in equation 15 is **not equal to zero**, which is required for the following proof of Theorem 2.
> >
> > Do I miss something?

---

> > > ### Author Response · Authors · 2021-11-22
> > > **Thank you for pointing out the typo**
> > >
> > > Thank you! That's a typo in the notation. Both $v$ for state $s'$ and $s$ should be defined exactly the same, i.e., w.r.t. $\tilde{\pi}_1$, $\tilde{\pi}_2$, , ..., $\tilde{\pi}_{j-1}$, $\pi_j$, ..., $\pi_N$. We corrected that typo in the latest manuscript and the rebuttal. Please let us know if there are any other questions.

---

> > > > ### Comment · Reviewer_YJEa · 2021-11-22
> > > > **Reply**
> > > >
> > > > What is the exact meaning of 'marginalizing $v(s^\prime_i)$ with respect to $\tilde{\pi}_1,\cdots,\tilde{\pi}_{j-1}$'?
> > > >
> > > > Why is $v(s^\prime_i)$ equal to $v_{\tilde{\pi}_1,\cdots,\tilde{\pi}_{j-1},\pi_{j},\cdots,\pi_N}(s^\prime_i)$?
> > > >
> > > > In my opinion, $v(s^\prime_i)$ is irrelevant to actions so it will not be changed after marginalizing with respect to policies. I think it is better to include the definitions of $v_{\tilde{\pi}_1,\cdots,\tilde{\pi}_{j-1},\pi_{j},\cdots,\pi_N}$ and $p_{\tilde{\pi}_1,\cdots,\tilde{\pi}_{j-1},\pi_{j},\cdots,\pi_N}(s_i^\prime|s_i,a_i)$ in terms of transition probabilities and policies.

---

> > > > > ### Author Response · Authors · 2021-11-22
> > > > > **Thank you for the question!**
> > > > >
> > > > > Thank you for raising the question!
> > > > >
> > > > > Marginalizing $v(s^\prime_i)$ w.r.t \tilde{\pi}_1, ...,\tilde{\pi}_{j-1} means the value of $s_i^\prime$, i.e., $v(s^\prime_i)$, is computed after $\pi_1$, ..., $\pi_{j-1}$ have been updated to $\tilde{pi}_1$, ..., $\tilde{\pi}_{j-1}$. So the transition probability is P_{$\tilde{\pi}_1, ..., \tilde{\pi}_{j-1}, \pi_j, ..., \pi_N$, and the policies at this moment are $\tilde{\pi}_1, ..., \tilde{\pi}_{j-1}, \pi_j, ..., \pi_N$, as specified in the paper.
> > > > >
> > > > > Why it should be this way? Since whenever one agent's policy is changed, the state transition probability is changed and the state distribution of $s_i'$ would be changed as well. One would thus need to condition $v(s_i')$ on the policies that have been updated. So $v(s_i^\prime)$ does change.

---

> > > > > > ### Comment · Reviewer_YJEa · 2021-11-22
> > > > > > **Reply**
> > > > > >
> > > > > > Could you tell me exactly why equation 10 is equal to equation 11?

---

> > > > > > > ### Author Response · Authors · 2021-11-22
> > > > > > > **Thank you for the thought-provoking discussion!**
> > > > > > >
> > > > > > > Thank you for the thought-provoking question!
> > > > > > >
> > > > > > > Yes, you were right. The $v(s_i^\prime)$ in equation 11 should be irrelevant to actions so it will not be changed, as also indicated by the matrix notation $G^{s_i}\rho_0$.
> > > > > > >
> > > > > > > We were confused by ourselves with the two different "value" functions: the true value function $v(s_i^\prime)$ and the baseline function as in Equation 13 (used to define the advantage). The baseline function $v(s_i)$ is defined differently from single-agent RL and should be conditioned on the transition dynamics and the current policies (such that the $L(\pi_j)=0$). We updated the manuscript and rebuttal to clarify this.
> > > > > > >
> > > > > > > Again, we want to emphasize the underlying idea: we want to define the advantage function for each agent such that the improvement guarantee emerges as long as this advantage is bounded (regardless its non-stationarity).

---

### Official Review · Reviewer_7r77 · 2021-11-03

**Correctness:** 4
**Technical Novelty And Significance:** 3
**Empirical Novelty And Significance:** 3
**Recommendation:** 6
**Confidence:** 2

**Main Review:**

The theory explains some of the success (monotonic improvement) of IPPO and MAPPO. The motivation is clear, derivations are easy to follow, and is to the best of my knowledge, novel and non-trivial.

However, while I like the theory presented, I am a little more skeptical with the impact of this paper, beyond reminding practitioners to decrease clipping ranges when there are more agents. I have a few minor gripes/questions
1) Section 6.3 seems fairly out of place and not in line with the central message of this paper (which is about monotonic improvement, as opposed to being about IPPO or MAPPO specifically).

2) Minor presentation issues. (i) Axis and labels on graphs are way too small (ii) Indices of summation can be explicitly stated for clarity. (iii) In section 6.1, is there a particular reason for using u^k as opposed to a^k? (iv) Why is the index k’ used in the definition of the surrogate objective and how is this different from k? (v) It would be very helpful to explain a little about the abbreviations used in SMAC (e.g., 10m vs 11m) etc, especially since these involve the number of agents (which is the key focus of this paper).

3) I found the experiments section quite difficult to follow. The first set of experiments (clipping and ratio changes) shows that clipping roughly bounds independent ratios if *hyperparameters are properly set*. These hyperparameters include optimization epochs and clipping range. It is not clear what “properly” means here. Further, I think it would be helpful to explicitly state why this disagreement with theory occurs.
4) The total variation reported in the second section of the experiments are based on samples collected by the behavior policy. Is this a good surrogate for the “true” TV between policies?

5) In the last experimental section, the authors report that “as the number of agents increases … empirical returns drop from nearly 20.0 to 17.5”, presumably to show that having too large a trust region (due to there being too many agents) would negatively impact performance. This comparison does not seem fair, since these are 2 very different scenarios and perhaps, even the optimal policy would do worse in the latter case. Maybe a better comparison would be to compare clipping over *joint* policies (effectively fewer agents), but in the same environment, to decentralized policies.


**Summary Of The Paper:**

This paper provides analysis showing monotonic improvements when in cooperative MARL settings, where independent, as opposed to joint ratios (over agents) are used. From a single agent’s perspective, a non-stationary distribution is experienced since other players are simultaneously performing updates to their policies, breaking the standard monotonic improvement property in TRPO. The authors’ propose a surrogate objective for decentralized policies, and show that the improvement in expected return can be bounded in terms of the sum of these (decentralized) surrogate objectives and a term proportionate to the total variation divergence between the 2 policies---analogous to the result by Schulman et al in 2005. Similarly, the authors proceed by showing that the TV divergence between policies can be bounded by enforcing constraints on independent ratios, where the strictness of these bounds depends on the number of agents. The trust region constraint is then split (independently) over all agents, and the authors show that prior methods like IPPO and MAPPO maximize an objective consistent with the proposed theory.

**Summary Of The Review:**

I recommend this paper for acceptance. The work is novel and does not have any glaring faults apart from some of my (possibly to-be-clarified) doubts with the experiments.

---

> ### Author Response · Authors · 2021-11-18
> **Thank you for the review**
>
> Thank you for raising these questions. We address them as follows:
>
> **Skeptical about the impact**:
> As noted by Reviewer YJEa,
> "decentralized policy learning with monotonic joint policy improvement is a very important problem for MARL".
> So the key result of our paper, i.e., a monotonic policy improvement for MARL, directly sheds light on this important problem.
> Furthermore, instead of ``reminding practitioners to decrease clipping ranges when there are more agents'',
> our paper shows, more importantly, that why and how this rato clipping works in theory and practice,
> which remained largely unclear for proximal methods before our paper, especially in MARL.
>
> **Q1**:
> Section 6.3 is intended to illustrate that IPPO and MAPPO are two instances of our monotonic improvement theory,
> despite their different ways of learning critics.
> It is to highlight that the trust region constraint is more crucial for learning policies,
> than the centralized or decentralized learning of critics. Both of these algorithms have recently empirically demonstrated state of the art performance on the SMAC benchmark tasks. By connecting these recent empirical results with this theory, our paper contributes deeper insight into how this empirical result was achieved.
>
> **Q2**:
> We will enlarge the labels and legends for all plots.
> We will make the summation indices explicit and the notations consistent.
> $k^\prime$ is used in the surrogate objective to mean that
> the surrogate objective is defined slightly differently for each agent,
> see the derivation in appendix, section A.3.2 on page 14, for details.
> We will also add the number of agents for each SMAC map in the figure caption.
>
> **Q3**:
> Thank you for raising this issue in the clarity of our results presentation.
> We think there is a minor misunderstanding here and hope to clarify it in discussion with the reviewer before updating the paper.
> There is no disagreement between the theory and the first set of experimental results.
> The theory requires the ratio to be bounded.
> Clipping is one of many ways to implement this but it is a heuristic approximation
> so often fails to bound ratios exactly within the ranges.
> However, clipping works in practice
> if the clipping range and the number of epochs are both well tuned ``properly''.
>
>
> **Q4**:
> We acknowledge that using samples generated by behavior policies
> may not estimate the \textit{true} divergence between two policies.
> Ideally, the TV should be computed over the whole state-action space
> with a sufficiently large number of randomly generated samples. However, this is intractable in practice so many studies, including the original publications on TRPO and PPO, resort to using the same off-policy samples we use instead.
> We will discuss this in the revised paper.
>
>
> **Q5**:
> Thank you for the suggestion.
> Yes, the optimal policy for these two maps may be different.
> We will remove the statement ``empirical returns drop from nearly $20.0$ to $17.5$'' and
> add the comparison between clipping over joint ratios and clipping over independent ones illustrated in the new figure below.
> Specifically, we apply the same clipping values to these two types of clipping,
> and use maps with many agents, i.e., 10m\_vs\_11m and 27m\_vs\_30m,
> to make the difference more salient (based on the theoretical results in the paper).
> The results are presented in section A.6 in the updated manuscript
> (can also be found in the supplementary materials).
> Compared to joint ratio clipping, the independent ratio clipping is
> more sensitive to the number of agents.
> In particular, for a small clipping value, e.g., $\epsilon=0.1$,
> joint ratio clipping consistently produces better performance than independent ratio clipping,
> even when the number of agents changes from 10 to 27.
> As the clipping value increases to $0.5$, the performance gap between these two types of clipping becomes larger,
> which is also aligned with our theoretical analysis.

---

### Official Review · Reviewer_JmGx · 2021-11-03

**Correctness:** 3
**Technical Novelty And Significance:** 3
**Empirical Novelty And Significance:** 2
**Recommendation:** 6
**Confidence:** 3

**Main Review:**

Strength:

1. The paper is written clearly, and the technique is relatively straightforward.
2. The analysis of ratio constraints between MAPPO and IPPO is interesting.

Weakness:

1. The paper considers the fully observable and cooperative case, which means all the agents have the same goal based on the same information. Therefore, it seems trivial to obtain the key result of the paper "bounding independent ratios based on the number of agents". Maybe the authors can further discuss the more general cases, like partial observation or competitive cases.
2. The authors compared the MAPPO and IPPO on different SMAC. I think more experiments and baselines are required to verify the results. At least, the  JR-PPO should be included to show the differences between joint ratio and independent ratios.

**Summary Of The Paper:**

This paper investigates how to enforce the trust-region constraint by bounding independent ratios based on the number of agents for cooperative multi-agent reinforcement learning. The authors also show that the surrogate objectives of IPPO and MAPPO are essentially equivalent when their critics converge to a fixed point, which is also supported by the empirical results.

**Summary Of The Review:**

In summary, I think this paper has some interesting results but can be further improved on both theoretical and experimental sides.

---

> ### Author Response · Authors · 2021-11-18
> **Thank you for the review!**
>
> Thank you for the review!
>
> **"Key result of the paper..."**:
> The ``bounding independent ratios based on the number of agents'' is a special instance of this improvement guarantee,
> for which we assume all agents share policy parameters and the trust region constraint can thus be delegated to each agent.
> Our analysis, e.g., Theorem 2, also naturally applies to more general cases where agents could have heterogenous state-action space
> and bounding independent ratios should then depend on state-action space of each agent, rather than simply the number of agents.
> Furthermore, our theoretical analysis sheds light on how ratio bounding would enforce a trust region constraint,
> and why/how ratio clipping works in practice,
> which is essentially important to understand the application of proximal methods in MARL.
> We will elaborate on this in the revised version.
>
>
> **Partial Observability**:
> Fully cooperative MARL does not mean that each agent has the full state information.
> On the contrary, each agent still has only its local state-action trajectories and the agent's policy is completely decentralized.
> Furthermore, the empirical results on SMAC included in the paper are in a partially observable setting,
> which also corroborates our theoretical analysis.
>
> **Competitive Games**:
> We respectfully disagree that an extension to competitive cases is necessary. The fully competitive setting already has a well established line of research, whilst the fully collaborative setting has recently been emphasised by the community [1,2] as a relatively underdeveloped topic of high importance for further research.
>
> [1] Dafoe, A., Bachrach, Y., Hadfield, G., Horvitz, E., Larson, K. and Graepel, T., Nature 2021. Cooperative AI: machines must learn to find common ground.
>
> [2] Zeynep Akata, et al. "A research agenda for hybrid intelligence: augmenting human intellect with collaborative, adaptive, responsible, and explainable artificial intelligence." Computer 53.08 (2020)
>
> **More Experiments**:
> We also report the JR-PPO results in section A.6 in the updated manuscript
> (can also be found in the supplementary materials),
> in which the joint ratio clipping is compared against the independent ratio clipping.
> Particularly, compared to joint ratio clipping, the independent ratio clipping is
> more sensitive to the number of agents.
> In particular, for a small clipping value, e.g., $\epsilon=0.1$,
> joint ratio clipping consistently produces better performance than independent ratio clipping,
> even when the number of agents changes from 10 to 27.
> As the clipping value increases to $0.5$, the performance gap between these two types of clipping becomes larger,
> which is also aligned with our theoretical analysis.

---

### Official Review · Reviewer_ax1X · 2021-11-03

**Correctness:** 3
**Technical Novelty And Significance:** 3
**Empirical Novelty And Significance:** 3
**Recommendation:** 8
**Confidence:** 3

**Main Review:**

Improvements and Questions:
- 0.1 Is a pretty standard clipping value, higher values are frequently not used for the reasons pointed out in this paper. I would have been very curious to see an ablation in the opposite direction, towards values < 0.1 since presumably the trust regions need to be smaller as the number of agents increase. Note, I am not suggesting that I would raise my score if this was done or requesting extra experiments, only that this would make for a better paper.
- Proposition 4 and the subsequent discussion is slightly confusing to me; I accept that both admit unique fixed points under appropriate assumptions and so estimate the advantage identically at that fixed point. However, the assumption underlying the use of extra information is to make the learning of the value function easier; is the implication here that there are not effects of centralized value functions on the learning?
- In Fig. 1d I find it surprising that the number of epochs appears to barely affect the total variational distance. This seems to suggest in turn that a large number of epochs should accelerate learning when the clipping ratio is 0.1; did you observe this?
- What values are the number of epochs fixed to in Fig. 1b and c?
- How does policies usually being recurrent in these settings factor into your results?

Clarity:
- The x axis on most of the SMAC graphs should be in scientific notation.
- The y axes on the SMAC graphs should probably not have underscores.
- In the legend of Fig. 4 and 5, it’s not clear what the (4) is in all the legends.
- Section 6.2 is written slightly confusingly in that it essentially says “a sufficient condition can be constructed to constrain independent ratios” and then follows up to say “ratio clipping is not sufficient to constrain independent ratios.” I think I understood what the author mean, that the estimation isn’t perfect so the constraint is not maintained, but I want to draw their attention to it having been confusing as a slight rewriting to clarify why the sufficient condition is not sufficient might be helpful.

**Summary Of The Paper:**

This paper works on extending the monotonic improvement guarantees of PPO to the multi-agent setting.

**Summary Of The Review:**

A good paper that helps explain some of the recent results studying on-policy methods in MARL.

---

> ### Author Response · Authors · 2021-11-18
> **Thank you for the review**
>
> Thank you for the review!
>
> **Smaller clipping values**:
> Thank you for suggesting this ablation study.
> We present the ablation results for small clipping values in the section A.5 in the updated manuscript
> (can also be found in the supplementary materials).
> It is true that a small clipping value results in a small trust region,
> and thus small clipping values, e.g., $0.08$, $0.05$ and $0.03$,
> would be preferred for maps with a large number of agents,
> e.g., maps 10m\_vs\_11m (10 agents) and 27m\_vs\_30m (27 agents).
> However, when the clip value is too small, e.g., $\epsilon=0.01$ in maps with 5 and 8 agents,
> the resultant trust region is also small and
> the update step in each iteration can thus be too small to improve the policy.
> Thus, one would need to trade off between the trust region constraint,
> to ensure monotonic improvement,
> and the policy update step,
> to ensure a sufficient parameter update at each iteration.
> We will also add these results to the appendix in the revised version.
>
> **Centralized value function**:
> Yes, the use of extra information is to make the value learning easier.
> However, Proposition 4 does not imply that the extra information could have no impact on learning.
> As showed in the [1], the use of centralized critics or decentralized ones is a bias-variance trade off:
> the centralized critic provides unbiased and correct on-policy return estimates,
> while also introduce higher policy gradient variance than the decentralized critic in practice.
> Please refer to [1] for further details.
>
> [1] Xueguang Lyu, Yuchen Xiao, Brett Daley, and Christopher Amato.
> Contrasting centralized and decentralized critics in multi-agent reinforcement learning.
> arXiv preprint arXiv:2102.04402, 2021
>
> **Number of epochs**:
> Yes, this phenomenon occurs when the clipping value is small, e.g., $0.1$ in Figure 1d,
> Also, note that Figure 1d only shows the cumulative percentage of
> TV divergence in the first round of actor updates.
> As the policy optimization proceeds, the impact of the number of epochs on
> TV divergence may increase.
> One may need to tune the learning rate to combat this side-effect as used in the implementation of Proximal Policy Optimization.
> We will elaborate more on this in the revised version.
> In addition, the number of epochs in Fig. 1b and c are set to 10,
> as we found it is robust and requires no learning rate decay.
>
> **Recurrent policies**:
> We acknowledge that the theoretical analysis considers only DecMDPs with non-reccurent policies.
> However, in the empirical experiments,
> we used recurrent networks, i.e., LSTM, as the decentralized policy architecture
> to overcome any partial observability in SMAC.
> These empirical results included in the paper also corroborate our theoretical analysis in this more general setting.
> We will clarify this when presenting empirical results in the revised paper.
>
> **Writing issues**:
> We will address the notation issues in all plots.
> In Figure 4 \& 5, the number in the legends means the number of repeated runs.
> We will change the confusing 'sufficient' statement to:
> "one can thus impose a sufficient condition to constrain independent ratios $\lambda^k$
> such that $\lambda^k\in [1-\frac{\alpha}{N}, 1+\frac{\alpha}{N}]$,
> where $N$ is the number of agents in training.
> Clipping is one of many ways to achieve this sufficient condition
> but itself is a heuristic approximation so often fails to bound ratios exactly within the ranges.
> In practice, one would need to tune the the clipping range and the number of epochs
> so the ratios can be properly bounded."

---

### Decision · Program_Chairs · 2022-01-20

**Decision:**

Reject

**Comment:**

The submission cannot be accepted as there seems to be a mistake in the proof of the main contribution (Theorem 2).